# CoRTX: Contrastive Framework for Real-time Explanation

**Yu-Neng Chuang\*[1], Guanchu Wang\*[1], Fan Yang[1], Quan Zhou[2], Pushkar Tripathi[2], Xuanting Cai[2], and Xia Hu[1]**
[1]Rice University, [2]Meta Platforms, Inc.
```
{ynchuang, guanchu.wang, fyang, xia.hu}@rice.edu,
{quanz, pushkart, caixuanting}@fb.com
```

## Abstract

Recent advancements in explainable machine learning provide effective and faithful solutions for interpreting model behaviors. However, many explanation methods encounter efficiency issues, which largely limit their deployments in practical scenarios. Real-time explainer (RTX) frameworks have thus been proposed to accelerate the model explanation process by learning a one-feed-forward explainer. Existing RTX frameworks typically build the explainer under the supervised learning paradigm, which requires large amounts of explanation labels as the ground truth. Considering that accurate explanation labels are usually hard to obtain due to constrained computational resources and limited human efforts, effective explainer training is still challenging in practice. In this work, we propose a COntrastive Real-Time eXplanation (CoRTX) framework to learn the explanation-oriented representation and relieve the intensive dependence of explainer training on explanation labels. Specifically, we design a synthetic strategy to select positive and negative instances for the learning of explanation. Theoretical analysis show that our selection strategy can benefit the contrastive learning process on explanation tasks. Experimental results on three real-world datasets further demonstrate the efficiency and efficacy of our proposed CoRTX framework. Our source code is available at: `https://github.com/ynchuang/CoRTX-720`

## 1 Introduction

The remarkable progress in explainable machine learning (ML) significantly improves the model transparency to human beings (Du et al., 2019). However, applying explainable ML techniques to real-time scenarios remains to be a challenging task. Real-time systems typically require model explanation to be not only effective but also efficient (Stankovic et al., 1992). Due to the requirements from both stakeholders and social regulations (Goodman & Flaxman, 2017; Floridi, 2019), the efficient model explanation is necessary for the real-time ML systems, such as the controlling systems (Steel & Angwin, 2010), online recommender systems (Yang et al., 2018), and healthcare monitoring systems (Gao et al., 2017). Nevertheless, existing work on non-amortized explanation methods has high explanation latency, including LIME (Ribeiro et al., 2016), KernelSHAP (Lundberg & Lee, 2017). These methods rely on either multiple perturbations or backpropagation in deep neural networks (DNN) for deriving explanation (Covert & Lee, 2021; Liu et al., 2021), which is time-consuming and limited for deployment in real-time scenarios.

Real-time explainer (RTX) frameworks have thus been proposed to address such efficiency issues and provide effective explanations for real-time systems (Dabkowski & Gal, 2017; Jethani et al., 2021b). Specifically, RTX learns an overall explainer on the training set by using the ground-truth explanation labels obtained through either exact calculation or approximation. RTX provides the explanation for each local instance via a single feed-forward process. Existing efforts on RTX can be categorized into two lines of work. The first line (Schwab & Karlen, 2019; Jethani et al., 2021b; Covert et al., 2022) explicitly learns an explainer to minimize the estimation error regarding to the approximated explanation labels. The second line (Dabkowski & Gal, 2017; Chen et al., 2018; Kanehira & Harada, 2019) trains a feature mask generator subject to certain constraints on pre-defined label distribution.

---

\*These authors contributed equally to this work.

Despite the effectiveness of existing RTX frameworks, recent advancements still rely on the large amounts of explanation labels under the supervised learning paradigm. The computational cost of obtaining explanation labels is extremely high (Roth, 1988; Winter, 2002), which thereby limits the RTX's deployment in real-world scenarios.

To tackle the aforementioned challenges, we propose a COntrastive Real-Time eXplanation (CoRTX) framework based on the contrastive learning techniques. CoRTX aims to learn the latent explanation of each data instance without any ground-truth explanation label. The latent explanation of an instance is defined as a vector encoded with explanation information. Contrastive learning has been widely exploited for improving the learning processes of downstream tasks by providing well-pretrained representative embeddings (Arora et al., 2019; He et al., 2020). In particular, task-oriented selection strategies of positive and negative pairs (Chen et al., 2020; Khosla et al., 2020) can shape the representation properties through contrastive learning. Motivated by the such contrastive scheme, CoRTX develops an explanation-oriented contrastive framework to learn the latent explanation, with the goal of further fine-tuning an explanation head in the downstream tasks.

CoRTX learns the latent explanation to deal with the explanation tasks by minimizing the contrastive loss (Van den Oord et al., 2018). Specifically, CoRTX designs a synthetic positive and negative sampling strategy to learn the latent explanation. The obtained latent explanation can then be transformed to feature attribution or ranking by fine-tuning a corresponding explanation head using a tiny amount of explanation labels. Theoretical analysis and experimental results demonstrate that CoRTX can successfully provide the effective latent explanation for feature attribution and ranking tasks. Our contributions can be summarized as follows:

- CoRTX provides a contrastive framework for deriving latent explanation, which can effectively reduce the required amounts of explanation label;

- Theoretical analysis indicate that CoRTX can effectively learn the latent explanation over the training set and strictly bound the explanation error;

- Experimental results demonstrate that CoRTX can efficiently provide explanations to the target model, which is applicable to both tabular and image data.

## 2 PRELIMINARY

### 2.1 NOTATIONS

We consider an arbitrary target model $f(\cdot)$ to interpret. Let input feature be $\boldsymbol{x} = [x_1, \cdots, x_M] \in \mathcal{X}$, where $x_i$ denote the value of feature $i$ for $1 \leq i \leq M$. The contribution of each feature to the model output can be treated as a cooperative game on the feature set $\mathcal{X}$. Specifically, the preceding difference $f(\widetilde{\boldsymbol{x}}_{\mathcal{S} \cup \{i\}}) - f(\widetilde{\boldsymbol{x}}_{\mathcal{S}})$ indicates the contribution of feature $i$ under feature subset $\mathcal{S} \subseteq \mathcal{U} \setminus \{i\}$, where $\mathcal{U}$ is the entire feature space. The overall contribution of feature $i$ is formalized as the average preceding difference considering all possible feature subsets $\mathcal{S}$, which can be formally given by

$$\phi_i(\boldsymbol{x}) := \mathbb{E}_{\mathcal{S} \subseteq \mathcal{U} \setminus \{i\}} \left[ f(\widetilde{\boldsymbol{x}}_{\mathcal{S} \cup \{i\}}) - f(\widetilde{\boldsymbol{x}}_{\mathcal{S}}) \right], \tag{1}$$

where $\widetilde{\boldsymbol{x}}_{\mathcal{S}} = \mathbf{S} \odot \boldsymbol{x} + (\mathbf{1} - \mathbf{S}) \odot \boldsymbol{x}_r$ denotes the perturbed sample, $\mathbf{S} = \mathbf{1}_{\mathcal{S}} \in \{0, 1\}^M$ is a masking vector of $\mathcal{S}$, and $\boldsymbol{x}_r = \mathbb{E}[\boldsymbol{x} \,|\, \boldsymbol{x} \sim P(\boldsymbol{x})]$ denotes the reference values[*] from feature distribution $P(\boldsymbol{x})$. The computational complexity of Equation 1 grows exponentially with the feature number $M$, which cumbers its application to real-time scenarios. To this end, we propose an efficient explanation framework for real-time scenarios in this work.

### 2.2 REAL-TIME EXPLAINER

Different from the existing non-amortized explanation methods (Lundberg & Lee, 2017; Lomeli et al., 2019) that utilize local surrogate models for explaining data instances, RTX trains a global model to provide fast explanation via one feed-forward process. Compared with the existing methods, the advantages of RTX mainly lie in two folds: (1) faster explanation generation; and (2) more robust explanation derivation. Generally, existing RTXs attempt to learn the overall explanation distribution using two lines of methodologies, which are Shapley-sampling-based approaches (Wang et al., 2021; Jethani et al., 2021b; Covert et al., 2022) and feature-selection-based approaches (Chen et al., 2018;

---

[*]Other statistic measurement can also be adopted for generating the reference value.

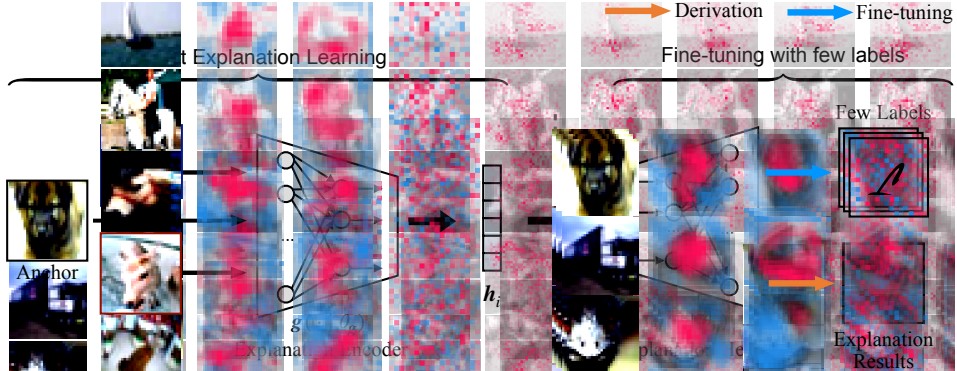

Figure 2: The explanation pipeline of the CoRTX framework, where $h_i$ denotes the latent explanation.

Dabkowski & Gal, 2017; Kanehira & Harada, 2019). The first line enforces the explainer to simulate a given approximated Shapley distribution for generating explanation results. A representative work, FastSHAP (Jethani et al., 2021b), exploits a DNN model to capture the Shapley distribution among training instances for efficient real-time explanations. The second line assumes the specific pre-defined feature patterns or distributions, and formulates the explainer learning process based on the given assumptions. One of the work (Chen et al., 2018) provides a feature masking generator for real-time feature selection. The training process of the mask generator is under the constraint of the given ground-truth label distribution. Both Shapley-sampling-based and feature-selection-based methods highly depend on explanation labels during the training phase. Although labels may bring strong supervised information to the explanation models that benefit explanation tasks, generating high-quality labels typically requires high computational time, which makes the labels hard to acquire. In this work, we propose an unsupervised learning paradigm CoRTX, which can significantly reduce the dependency on the explanation labels. The proposed CoRTX benefits the explanation tasks by exploiting a contrastive framework to generate latent explanations without relying on the labels.

## 2.3 LIMITATION OF SUPERVISED FRAMEWORK

Supervised RTX typically relies on large quantities of ground-truth explanation labels, which limits its application in practice. We experimentally show that supervised RTX suffers from the performance degradation when explanation labels are insufficient. Specifically, taking the Shapley value as the ground truth, supervised RTX learns to minimize the cross entropy on the training set, and then estimates the feature ranking on the testing set for evaluation. The results on Census dataset (Dua & Graff, 2017) demonstrate the intensive dependence of supervised RTX on the amount of explanation labels. The performance of feature ranking versus the exploitation ratio of explanation label is shown in Figure 1, where the implementation details are introduced in Appendix D. Here, we observe that the performance of feature ranking significantly enhances as the ratio of label increases. Supervised RTXs outperform FastSHAP, which is

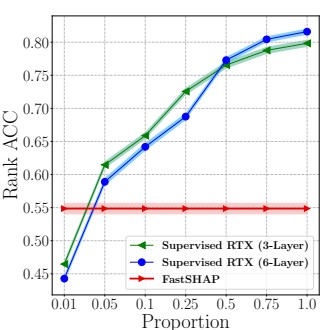

Figure 1: The Rank ACC v.s. the explanation label ratio in RTX.

trained with approximated labels, under 5% label usage. However, the complexity of calculating Shapley values generally grows exponentially with the feature number, which makes it challenging to train RTX with a large amount of labels. We propose CoRTX in this paper to train the explainer in a light-label manner, where only a small amount of explanation labels are used for fine-tuning the downstream head.

## 3 CONTRASTIVE REAL-TIME EXPLANATION

We systematically introduce the CoRTX framework in this section. Figure 2 illustrates the overall pipeline of CoRTX. In particular, our pipeline first uses an explanation encoder to learn the latent explanation in a contrastive way, and then an explanation head is further tuned with a small amount

of explanation labels. An explanation-oriented positive augmentation and negative sampling strategy is designed for the latent explanation learning process.

## 3.1 Positive Data Augmentation toward Similar Explanation

Different from the conventional data augmentation (Liu et al., 2021) for representation learning, CoRTX develops an explanation-oriented data augmentation strategy for training RTX, where the augmented positive and negative instances of explanation are proved to be beneficial (Kim et al., 2016; Dhurandhar et al., 2018). Specifically, considering an anchor data instance $\boldsymbol{x} \in \mathcal{X}$, the set of synthetic positive instances $\mathcal{X}^+$ is generated via $m$ times of independent perturbations on $\boldsymbol{x}$:

$$\mathcal{X}^+ = \{\mathbf{S}_i \odot \boldsymbol{x} + (\mathbf{1} - \mathbf{S}_i) \odot \boldsymbol{x}_r \mid \mathbf{S}_i \sim \mathcal{B}(M, \lambda), 1 \leq i \leq m\}, \tag{2}$$

where $\mathbf{S}_i$ is sampled from the $M$-dim binomial distribution, and $\lambda$ controls the probability density.

CoRTX provides the explanation-oriented data augmentation by selecting the compact positive instances $\widetilde{\boldsymbol{x}}^+$ from $\mathcal{X}^+$. Intuitively, $\widetilde{\boldsymbol{x}}^+$ is expect to obtain the most similar model explanation as the corresponding $\boldsymbol{x}$, which means the feature attribution between $\widetilde{\boldsymbol{x}}^+$ and $\boldsymbol{x}$ are highly similar. However, the greedy selection of $\widetilde{\boldsymbol{x}}^+$ needs the explanation score of each instance in $\mathcal{X}^+$, which is hard to access in practice. To tackle the issue, we propose Theorem 1 to guide the selection of $\widetilde{\boldsymbol{x}}^+$ through a set of synthetic positive instances without accessing the explanation scores. The proof of Theorems 1 is shown in Appendix A.

**Theorem 1** (**Compact Alignment**). *Let $f(\boldsymbol{x})$ be a $K_f$-Lipschitz* [†] *continuous function and $\boldsymbol{\Phi}(\boldsymbol{x}) = [\phi_1(\boldsymbol{x}), \cdots, \phi_M(\boldsymbol{x})]$ be the feature importance scores, where $\phi_i(\boldsymbol{x}) := \mathbb{E}_{\mathcal{S} \subseteq \mathcal{U} \backslash \{i\}} \big[ f(\widetilde{\boldsymbol{x}}_{\mathcal{S} \cup \{i\}}) - f(\widetilde{\boldsymbol{x}}_{\mathcal{S}}) \big]$. Given a perturbed positive instance $\widetilde{\boldsymbol{x}} \in \mathcal{X}^+$ satisfying $\min_{1 \leq i \leq M} \phi_i(\widetilde{\boldsymbol{x}}) \geq 0$, the explanation difference $\|\boldsymbol{\Phi}(\boldsymbol{x}) - \boldsymbol{\Phi}(\widetilde{\boldsymbol{x}})\|_2$ is bounded by the prediction difference $|f(\boldsymbol{x}) - f(\widetilde{\boldsymbol{x}})|$ as*

$$\|\boldsymbol{\Phi}(\boldsymbol{x}) - \boldsymbol{\Phi}(\widetilde{\boldsymbol{x}})\|_2 \leq (1 + \sqrt{2}\gamma_0)|f(\boldsymbol{x}) - f(\widetilde{\boldsymbol{x}})| + \sqrt{M}\gamma_0, \tag{3}$$

*where $\gamma_0 = K_f \|\boldsymbol{x}\|_2$ and $K_f \geq 0$ is the Lipschitz constant of function $f(\cdot)$.*

Theorem 1 basically shows that the explanation difference can be bounded by the prediction difference between the anchor $\boldsymbol{x}$ and the perturbed positive instance $\widetilde{\boldsymbol{x}}$. The compact positive instance $\widetilde{\boldsymbol{x}}^+$ in Theorem 1 can derive the minimal difference of model outputs and further result in a similar explanation to the anchor $\boldsymbol{x}_i$. Following Theorem 1, CoRTX selects $\widetilde{\boldsymbol{x}}_i^+ \in \mathcal{X}^+$ by holding the minimum preceding difference between of $f(\boldsymbol{x})$ and $f(\widetilde{\boldsymbol{x}}_i)$. Formally, the selection strategy can be indicated as:

$$\widetilde{\boldsymbol{x}}^+ = \arg \min_{\widetilde{\boldsymbol{x}}_i \in \mathcal{X}^+} |f(\boldsymbol{x}) - f(\widetilde{\boldsymbol{x}}_i)|, \tag{4}$$

where $f(\cdot)$ is the target model for explanation.

Selecting excessively positive instances has been proved to degrade contrastive learning (Zhu et al., 2021). The excessively positive instances refer to those very similar to the anchor instance. Regarding our $\widetilde{\boldsymbol{x}}^+$ may become excessively positive when selected from the universe set, $\mathcal{X}^+$ is set to be much smaller than the entire universe set, where $|\mathcal{X}^+| = m \ll 2^M$. For example, we set $|\mathcal{X}^+| = 300 \ll 2^{96}$ on the Bankruptcy dataset (Liang et al., 2016). In practice, CoRTX uses Equation 4 to search the compact positive instance $\widetilde{\boldsymbol{x}}^+$ from $\mathcal{X}^+$, where the excessively positive instances may be avoided. In this way, CoRTX can ensure hard positive instances to be as many as possible, which benefits the latent explanation learning process.

## 3.2 Explanation Contrastive Loss

The proposed CoRTX adopts the contrastive learning (He et al., 2020; Chen et al., 2020) to generate the latent explanation by training an explanation encoder $g(\cdot \mid \theta_g) : \mathbb{R}^M \to \mathbb{R}^d$. Here, a positive pair and several negative pairs are exploited during the training of $g(\cdot \mid \theta_g)$. A positive pair includes the anchor $\boldsymbol{x}_i$ and a compact positive instance $\widetilde{\boldsymbol{x}}_i^+$. A negative pair contains the anchor $\boldsymbol{x}_i$ and a randomly selected instance $\boldsymbol{x}_j$ ($j \neq i$), where the explanations are mostly different. Let $\boldsymbol{h}_i = g(\boldsymbol{x}_i \mid \theta_g)$, $\widetilde{\boldsymbol{h}}_i^+ = g(\widetilde{\boldsymbol{x}}_i^+ \mid \theta_g)$ be the latent explanation of the positive pair, and $\boldsymbol{h}_i$, $\boldsymbol{h}_j = g(\boldsymbol{x}_j \mid \theta_g)$ be the latent explanation for a negative pair. CoRTX trains the encoder $g(\cdot \mid \theta_g)$ by maximizing the dot similarity

---

[†]Lipschitz continuity can be refered to (Virmaux & Scaman, 2018).

of positive pairs and minimizing the dot similarity of negative pairs. Following the criterion, $g(\cdot \mid \theta_g)$ essentially minimizes the contrastive loss as follows:

$$\mathcal{L}_g = -\log \frac{\exp\left(\boldsymbol{h}_i \cdot \widetilde{\boldsymbol{h}}_i^+/\tau\right)}{\sum_{j=1}^N \exp\left(\boldsymbol{h}_i \cdot \boldsymbol{h}_j/\tau\right)}, \tag{5}$$

where $\tau$ is a temperature hyper-parameter (Wu et al., 2018) and $N$ denotes the batch size for training.

To provide the explanation for $f(\cdot)$ from the learned latent explanation in $g(\cdot \mid \theta_g)$, an explanation head $\boldsymbol{\eta}(\cdot \mid \theta_{\boldsymbol{\eta}}) : \mathbb{R}^d \to \mathbb{R}^M$ is further tuned based on the certain explanation tasks. Specifically, $\boldsymbol{\eta}(\cdot \mid \theta_{\boldsymbol{\eta}})$ is tuned with a small amount of explanation labels (e.g., the Shapley values), where the computational cost is relatively affordable in real-world settings. $\boldsymbol{\eta}(\cdot \mid \theta_{\boldsymbol{\eta}})$ is specifically designed to cope with the different scenarios of model explanation. According to the effects of Equation 4 and Equation 5, CoRTX enforces $\boldsymbol{h}_i$ to be similar to $\widetilde{\boldsymbol{h}}_i^+$ and dissimilar to $\boldsymbol{h}_j$, which is consistent with the relationship of their model explanations. This enables the learned representation to contain the explanation information. In this work, CoRTX is studied under two common explanation scenarios, i.e., feature attribution task and feature ranking task. To ensure the explanation quality of CoRTX, we here provide Theorem 2 to show that the explanation error of CoRTX is bounded with theoretical guarantees. The proof of Theorem 2 is provided in Appendix B.

**Theorem 2 (Explanation Error Bound).** *Let $\boldsymbol{\eta}(\cdot|\theta_{\boldsymbol{\eta}})$ be $K_\eta$-Lipschitz continuous, and $\hat{\boldsymbol{\Phi}}(\cdot) = \boldsymbol{\eta}(g(\cdot \mid \theta_g)|\theta_{\boldsymbol{\eta}})$ be the estimated explanations. If the encoder $g(\cdot|\theta_g)$ is well-trained, satisfying $\|\boldsymbol{\Phi}(\boldsymbol{x}) - \hat{\boldsymbol{\Phi}}(\boldsymbol{x})\|_2 \leq \mathcal{E} \in \mathbb{R}^+$, then the explanation error on testing instance $\boldsymbol{x}_k$ can be bounded as:*

$$\|\boldsymbol{\Phi}(\boldsymbol{x}_k) - \hat{\boldsymbol{\Phi}}(\boldsymbol{x}_k)\|_2 \leq (1+\sqrt{2}\gamma_k)|f(\boldsymbol{x}_k) - f(\widetilde{\boldsymbol{x}}_k^+)| + \sqrt{M}\gamma_k + \mathcal{E} + K_\eta\|\boldsymbol{h}_{\widetilde{\boldsymbol{x}}_k^+} - \boldsymbol{h}_{x_k}\|_2, \tag{6}$$

*where $\widetilde{\boldsymbol{x}}_k^+$ is a compact positive instance, $\boldsymbol{h}_{\boldsymbol{x}_k} = g(\boldsymbol{x}_k|\theta_g)$, and $\gamma_k = K_f\|\boldsymbol{x}_k\|_2$.*

Theorem 2 shows that the upper bound of explanation error depends on four different terms, which are $\gamma_k$, training error $\mathcal{E}$, $|f(\boldsymbol{x}) - f(\widetilde{\boldsymbol{x}}_k^+)|$, and $\|\boldsymbol{h}_{\widetilde{\boldsymbol{x}}_k} - \boldsymbol{h}_{\boldsymbol{x}_k}\|_2$. CoRTX contributes to minimize the upper bound by explicitly minimizing $|f(\boldsymbol{x}) - f(\widetilde{\boldsymbol{x}}_k^+)|$ and $\|\boldsymbol{h}_{\widetilde{\boldsymbol{x}}_k} - \boldsymbol{h}_{\boldsymbol{x}_k}\|_2$ as follows:

- CoRTX selects the compact positive instances following the selection strategy in Equation 4, which obtains the minimal value of $|f(\boldsymbol{x}) - f(\widetilde{\boldsymbol{x}}_k^+)|$;

- CoRTX maximizes the similarity between positive pairs while training the encoder $g(\cdot|\theta_g)$, which minimizes the value of $\|\boldsymbol{h}_{\widetilde{\boldsymbol{x}}_k} - \boldsymbol{h}_{\boldsymbol{x}_k}\|_2$.

Besides the two terms mentioned above, the error $\mathcal{E}$ is naturally minimized after $g(\cdot \mid \theta_g)$ is trained, and $\gamma_k$ keeps constant for the given $f(\cdot)$. Thus, we can theoretically guarantee the explanation error bound of CoRTX by Equation 6. This ensures the capability of CoRTX to provide effective explanations to the target model over the testing set.

## 3.3 Algorithm of CoRTX

The outline of CoRTX is given in Algorithm 1. CoRTX follows Equation 4 for compact positive selection (lines 4-5), and learns the explanation encoder $g(\cdot \mid \theta_g)$ according to Equation 5 (line 6). The training iteration terminates when $g(\cdot \mid \theta_g)$ is converged. Overall, $g(\cdot \mid \theta_g)$ learns the latent explanation for each instance $\boldsymbol{x} = [x_1, \cdots, x_M] \in \mathcal{X}$, and $\boldsymbol{\eta}(\cdot \mid \theta_{\boldsymbol{\eta}})$ is then fine-tuned with a small amount of explanation labels for particular tasks. The algorithmic process of CoRTX can be expressed as $\boldsymbol{\eta}(g(\boldsymbol{x}_i \mid \theta_g) \mid \theta_{\boldsymbol{\eta}})$ in general.

---

**Algorithm 1:** Real-Time Explainer Training with CoRTX

---

1   **Input:** Target model $f$ and input feature values $\boldsymbol{x} = [x_1, \cdots, x_M]$.

2   **Output:** Estimated explanation values of each feature $[\hat{\phi}_1, ..., \hat{\phi}_M]$.

3   **while** *not convergence* **do**

4      Generate the synthetic positive instances using $\mathcal{X}^+$ from Equation 2.

5      Select a compact positive instance $\widetilde{\boldsymbol{x}}^+$ by Equation 4 and the set of negative instances $\{\boldsymbol{x}_j \mid j \neq i\}$.

6      Update $\boldsymbol{h}_{\boldsymbol{x}} = g(\boldsymbol{x} \mid \theta_g)$ with $\widetilde{\boldsymbol{x}}^+$ and $\boldsymbol{x}_j$ to minimize loss function given by Equation 5.

7   **end**

8   Fine-tune $\boldsymbol{\eta}(\boldsymbol{h}_{\boldsymbol{x}} \mid \theta_{\boldsymbol{\eta}})$ with a small amount of explanation labels.

---

# 4 EXPERIMENTS

In this section, we conduct experiments to evaluate the performance of CoRTX, aiming to answer the following three research questions: **RQ1:** How does CoRTX perform on explanation tasks in terms of the efficacy and efficiency compared with state-of-the-art baselines? **RQ2:** Is the latent explanation from CoRTX effective on fine-tuning the explanation head? **RQ3:** Does synthetic augmentation contribute to the explanation performance of CoRTX?

## 4.1 DATASETS AND BASELINES

**Datasets.** Our experiments consider two tabular datasets: Census (Dua & Graff, 2017) with 13 features, Bankruptcy (Liang et al., 2016) with 96 features, and one image dataset: CIFAR-10 (Krizhevsky et al., 2009) with $32 \times 32$ pixels. The preprocessing and statistics of three datasets are provided in Appendix C. **Baseline Methods.** In Census and Bankruptcy datasets, CoRTX is compared with two RTX methods, i.e., Supervised RTX and FastSHAP (Jethani et al., 2021b), as well as two non-amortized explanation methods, i.e., KernelSHAP (KS) (Lundberg & Lee, 2017) and Permutation Sampling (PS) (Mitchell et al., 2021). In CIFAR-10 dataset, CoRTX is compared with FastSHAP and other non-amortized methods, including DeepSHAP (Lundberg & Lee, 2017), Saliency (Simonyan et al., 2013), Integrated Gradients (IG) (Sundararajan et al., 2017), SmoothGrad (Smilkov et al., 2017), and GradCAM (Selvaraju et al., 2017). More details about the baselines can be found in Appendix D.

## 4.2 EXPERIMENTAL SETTINGS AND EVALUATION METRICS

In this part, we introduce the experimental settings and metrics for evaluating CoRTX. The considered explanation tasks and implementation details are shown as follows.

**Feature Attribution Task.** This task aims to test the explanation performance on feature attribution. We here implement CoRTX-MSE on fine-tuning the explanation head $\boldsymbol{\eta}(\boldsymbol{x}_i \mid \theta_{\boldsymbol{\eta}})$. Given $g(\boldsymbol{x}_i \mid \theta_g)$, the explanation values are predicted through $[\hat{\phi}_1, \cdots, \hat{\phi}_M] = \boldsymbol{\eta}(g(\boldsymbol{x}_i \mid \theta_g) \mid \theta_{\boldsymbol{\eta}})$, where $\hat{\phi}_i$ indicates the attribution scores of feature $i$. Let $[\phi_1, \cdots, \phi_M]$ be the explanation label of an instance $\boldsymbol{x}_i$, CoRTX-MSE learns $\boldsymbol{\eta}(\cdot \mid \theta_{\boldsymbol{\eta}})$ by minimizing the mean-square loss $\mathcal{L}_{\text{MSE}} = \frac{1}{M} \sum_{j=1}^{M} (\hat{\phi}_j - \phi_j)^2$. To evaluate the performance, we follow the existing work (Jethani et al., 2021b) to estimate the $\ell_2$-error of each instance on the testing set, where the $\ell_2$-error is calculated by $\sqrt{\sum_{j=1}^{M} (\phi_j - \hat{\phi}_j)^2}$.

**Feature Importance Ranking Task.** This task aims to evaluate the explanation performance on feature ranking index. We here implement CoRTX-CE for fine-tuning the explanation head $\boldsymbol{\eta}(\cdot \mid \theta_{\boldsymbol{\eta}})$. Given $g(\boldsymbol{x}_i \mid \theta_g)$, the feature ranking index is generated by $[\hat{\mathbf{r}}_1, \cdots, \hat{\mathbf{r}}_M] = \boldsymbol{\eta}(g(\boldsymbol{x}_i \mid \theta_g) \mid \theta_{\boldsymbol{\eta}})$ for each instance $\boldsymbol{x}_i$. The predicted feature ranking index $[\hat{r}_1, \cdots, \hat{r}_M]$ is given by $\hat{r}_j = \arg\max \hat{\mathbf{r}}_j$ for $1 \leq j \leq M$, where $\arg\max(\cdot)$ returns the index with the maximal importance score. Let $[r_1, \cdots, r_M]^{\ddagger}$ be the ground-truth ranking label, CoRTX-CE learns $\boldsymbol{\eta}(\cdot \mid \theta_{\boldsymbol{\eta}})$ by minimizing the loss function $\mathcal{L}_{\text{CE}} = \sum_{j=1}^{M} \text{CrossEntropy}(\hat{\mathbf{r}}_j, \texttt{onehot}(r_j))$. To evaluate the feature importance ranking, we follow existing work (Wang et al., 2022), where the ranking accuracy of each instance is given by Rank ACC $= (\sum_{j=1}^{M} \frac{\mathbf{1}_{\hat{r}_j = r_j}}{j}) / (\sum_{j=1}^{M} \frac{1}{j})$.

**Evaluation of Efficiency.** The algorithmic throughput is exploited to evaluate the speed of explaining process (Wang et al., 2022; Teich & Teich, 2018). Specifically, the throughput is calculated by $\frac{N_{\text{test}}}{t_{\text{total}}}$, where $N_{\text{test}}$ and $t_{\text{total}}$ denote the testing instance number and the overall time consumption of explanation derivation, respectively. Higher throughput indicates a higher efficiency of the explanation process. In experiments, $t_{\text{total}}$ is measured based on the physical computing infrastructure given in Appendix E.3.

**Implementation Details.** In experiments, CoRTX is implemented under three different prediction models $f(\cdot)$. AutoInt (Song et al., 2019) is adopted for Census data, MLP model is used for Bankruptcy data, and ResNet-18 is employed for CIFAR-10. As for the ground-truth explanation labels, we calculate the exact Shapley values for Census dataset. For the other two datasets, we utilize the approximated methods to generate explanation labels (Covert & Lee, 2021; Jethani et al., 2021b), due to the high time complexity of exact Shapley value calculation. The estimated values

---

‡The ranking labels are sorted by the ground-truth explanation scores (e.g., exact or approximated Shapley values.)

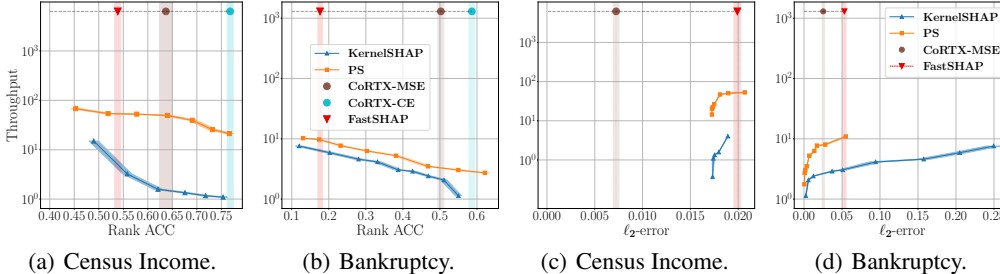

Figure 3: Explanation throughput versus ranking accuracy on Census Income (a) and Bankruptcy dataset (b). Explanation throughput versus $\ell_2$-error on Census Income (c) and Bankruptcy dataset (d).

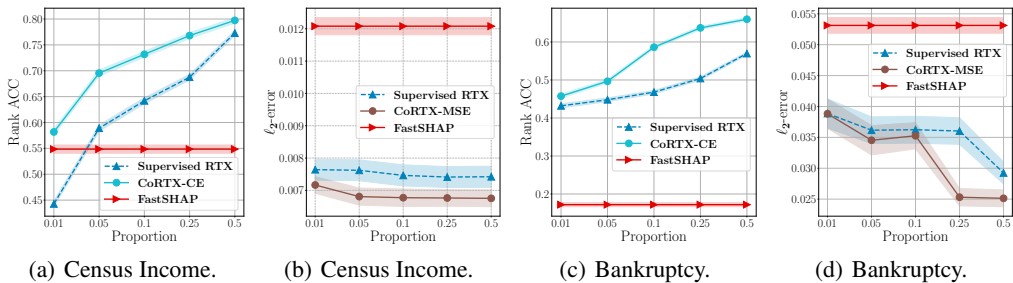

Figure 4: Explanation performance with different proportions of explanation label usage on two tabular datasets. CoRTX outperforms SOTA RTX baselines by using only 5% of labels.

from Antithetical Permutation Sampling (APS) (Mitchell et al., 2021; Lomeli et al., 2019) and KS are proved to be infinitely close to the exact Shapley values (Covert & Lee, 2021) when they involve sufficient samples. In experiments, we employ APS on Bankruptcy and KS on CIFAR-10 for obtaining explanation labels. More details are introduced in Appendix E.

## 4.3 Tabular Experiments

### 4.3.1 Explanation Efficacy and Efficiency (RQ1)

We compare CoRTX with the baseline methods on deriving model explanations. The results in feature attribution and feature ranking are shown by $\ell_2$-error versus throughput and Rank ACC versus throughput, respectively. The reported results of CoRTX adopts 25% of the explanation labels on fine-tuning the explanation head. The performance comparisons are illustrated in Figures 3, where (a), (b) demonstrate the performance on feature ranking and (c), (d) show the performance on feature attribution. According to the experimental results, we have the following observations:

- **CoRTX vs. Non-Amortized Methods**: Compared with the non-amortized explanation methods with large times of model evaluations, CoRTX achieves competitive explanation performance. In Census data, we observe CoRTX is competitive to PS and KS with respectively $2^9$ and $2^{10}$ of model evaluations. For KS and PS, it is noted that there is a significant decrease in Rank ACC and an obvious increase in $\ell_2$-error as the throughput grows. This indicates that KS and PS both suffer from an undesirable trade-off between the explanation speed and performance. In contrast, the proposed CoRTX achieves both efficient and effective explanations for the target model.

- **CoRTX vs. FastSHAP**: CoRTX outperforms FastSHAP on Rank ACC and $\ell_2$-error under the same level of throughput. This observation shows that CoRTX provides more effective explanation under the scenario of RTX.

- **CoRTX-MSE vs. CoRTX-CE**: CoRTX consistently provides effective solutions for the two explanation tasks. CoRTX-MSE is competitive on the feature attribution and ranking task, while CoRTX-CE performs even better than CoRTX-MSE in the ranking task. The superiority of CoRTX-CE on ranking task largely results from the loss function in fine-tuning, where appropriate tuning loss can help enhance the explanation quality for certain tasks.

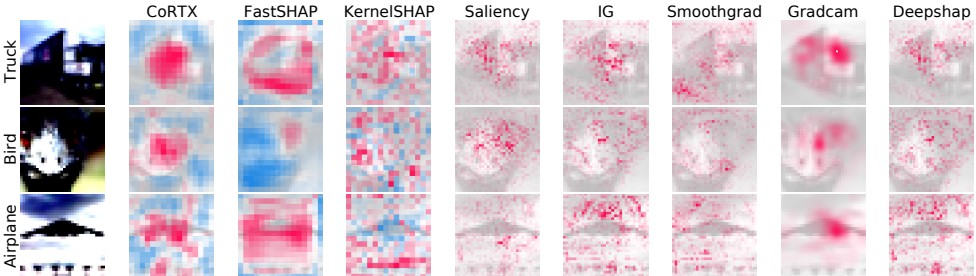

Figure 5: Explanations generated on CIFAR-10 Dataset.

### 4.3.2 CONTRIBUTIONS ON LATENT EXPLANATION (RQ2)

In this experiment, we study the effectiveness of latent explanation in deriving explanation results. Given the obtained latent explanation, Figure 4 demonstrates the explanation performance of the RTX frameworks with different label usage ratios. We compare CoRTX with FastSHAP and Supervised RTX and summarize the key observations as below:

- **Effectiveness of Encoding**: CoRTX-MSE and CoRTX-CE consistently outperform the Supervised RTX on different label proportions. The results on two explanation tasks show that CoRTX provides effective latent explanation $h_i$ in fine-tuning the head $\eta(h_i \mid \theta_\eta)$.
- **Sparsity of Labels**: CoRTX-MSE and CoRTX-CE can provide effective explanation outcomes when explanation labels are limited. The results indicate that CoRTX can be potentially applied to large-scale datasets, where the explanation labels are computationally expensive to obtain.
- **Effectiveness of Labels**: CoRTX and Supervised RTX consistently outperform FastSHAP when having more than $5\%$ explanation labels. FastSHAP synchronously generates the explanation labels with low-sampling times during the training. In contrast, CoRTX and Supervised RTX are trained with high-quality labels, which are the exact Shapley values or the approximated ones with high-sampling times. The results indicate that high-quality explanation labels can significantly benefit the learning process of the explainer, even when exploiting only $5\%$ explanation labels.

### 4.3.3 ABLATION STUDIES ON SYNTHETIC POSITIVE AUGMENTATIONS (RQ3)

We further conduct the ablation studies on synthetic positive augmentation in Section 3.1, aiming to prove its validity in learning latent explanation. Since negative pairs are randomly selected in CoRTX, we here mainly focus on the positive augmentation in our studies. The ablation studies are conducted under both attribution and ranking tasks on Census dataset. In the experiments, CoRTX is compared with two methods: *CoRTX w/o Compact Alignment (CA)* and *CoRTX w/ Maximum Alignment (MA)*. To be concrete, *CoRTX w/o CA* replaces Equation 4 with the existing settings of contrastive learning, which is a random masking strategy. *CoRTX w/ MA* replaces the minimum selecting strategy in Equation 4 with the maximum one on synthetic instance set, which is randomly masked from the anchor instance. The remaining settings follow the same as CoRTX. Figure 6 demonstrates the comparisons among *CoRTX w/o CA*, *CoRTX w/ MA* and CoRTX. We can observe that CoRTX outperforms *CoRTX w/o CA* and *CoRTX w/ MA* under different proportions of explanation label

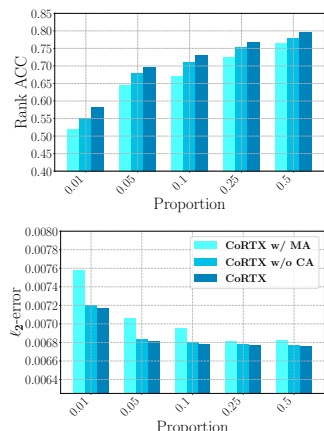

Figure 6: Ablation Study on Positive Augmentations.

usage. This demonstrates that the designed synthetic positive augmentation significantly benefits the efficacy of explanation results.

### 4.4 IMAGE EXPERIMENTS (RQ1)

Unlike the tabular cases, image data is typically composed of high-dimension pixel features. To further test the efficacy of latent explanation, we evaluate CoRTX on CIFAR-10 dataset. We compare CoRTX with FastSHAP and six non-amortized image attribution methods. The explanation results of CoRTX are fine-tuned using $5\%$ of ground-truth explanation labels. CoRTX and FastSHAP output

|  | Top-1 Accuracy | | Log-odds | |
|---|---|---|---|---|
|  | Exclusion | Inclusion | Exclusion | Inclusion |
| CoRTX | **0.373** ± 0.011 | **0.774** ± 0.012 | **5.172** ± 0.010 | **4.863** ± 0.028 |
| FastSHAP | 0.420 ± 0.015 | **0.782** ± 0.015 | **5.010** ± 0.012 | 4.817 ± 0.026 |
| KernelSHAP | 0.395 ± 0.012 | 0.769 ± 0.012 | 5.517 ± 0.009 | 4.829 ± 0.024 |
| Saliency | 0.499 ± 0.012 | 0.735 ± 0.013 | 5.869 ± 0.010 | 4.415 ± 0.025 |
| IG | 0.559 ± 0.012 | 0.740 ± 0.012 | 6.159 ± 0.011 | 4.439 ± 0.025 |
| Smoothgrad | 0.471 ± 0.011 | 0.738 ± 0.012 | 5.679 ± 0.008 | 4.370 ± 0.026 |
| Gradcam | 0.563 ± 0.012 | 0.741 ± 0.012 | 6.109 ± 0.011 | 4.446 ± 0.024 |
| Deepshap | 0.554 ± 0.012 | 0.742 ± 0.012 | 6.124 ± 0.009 | 4.441 ± 0.023 |

Table 1: Exclusion and Inclusion AUCs. The evaluation scores are calculated from the bootstrapping average scores of 20 times repetitions. The explanation methods perform better when obtaining lower Exclusion AUC and encountering higher Inclusion AUC on both Top-1 Accuracy and Log-odds.

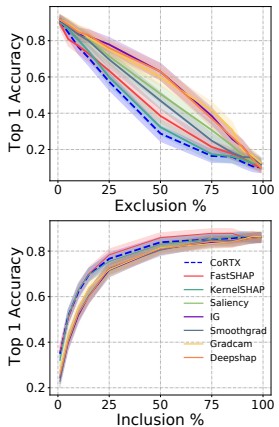

Figure 7: AUC on CIFAR-10

2×2 super-pixel attributions (Jethani et al., 2021b) for explaining the prediction of image classification. More details are provided in Appendix E.

### 4.4.1 A Case Study of CoRTX

We visualize the explanation results in Figure 5. It shows that CoRTX, FastSHAP, and GradCAM are able to highlight the relevant objects corresponding to the image attribution labels. Specifically, CoRTX provides better human-understandable explanation results. In contrast, we observe that the important regions localized by FastSHAP are larger than relevant objects. Moreover, GradCAM highlights the regions only on tiny relevant objects. This makes FastSHAP and GradCAM less human-understandable for the explanation. These observations validate that the explanations from CoRTX are more human-understandable. More case studies are available in Appendix H.

### 4.4.2 Quantitative Evaluation

To quantitatively investigate the explanation results on CIFAR-10, we test CoRTX with the exclusion and inclusion comparing with FastSHAP as well as Shapley-based and gradient-based non-amortized methods. Following the experimental settings from existing work (Petsiuk et al., 2018; Jethani et al., 2021b), we utilize the AUC of Top-1 accuracy and the AUC of Log-odds as the metrics for evaluation (Jethani et al., 2021b). Specifically, the testing images are orderly masked according to the estimated feature importance scores. Once the important pixels are removed from the input images, Top-1 accuracy and Log-odds are expected to drop drastically, obtaining a low exclusion AUC and a high inclusion AUC. More details about Log-odds are given in Appendix G. Figure 6 and Table 1 demonstrate the results of the exclusion and inclusion under 1000 images. We observe that CoRTX outperforms all other baselines in two evaluations, which obtains the lowest exclusion AUC of Top-1 Accuracy and the lowest inclusion AUC of Log-odds. CoRTX is also competitive with the state-of-the-art baseline, FastSHAP, on the other two evaluations. Besides, CoRTX performs better than non-amortized baseline methods in all four evaluations.

## 5 Conclusions

In this paper, we propose a real-time explanation framework, CoRTX, based on contrastive learning techniques. Specifically, CoRTX contains two components, i.e., explanation encoder and explanation head, which learns the latent explanation and then provides explanation results for certain explanation tasks. CoRTX introduces an explanation-oriented data augmentation strategy for learning the latent explanation. An explanation head can be further fine-tuned with a small amount of explanation labels. Theoretical analysis indicates our proposed data augmentation and contrastive scheme can effectively bind the explanation error. The experimental results on three datasets demonstrate that CoRTX outperforms the state-of-the-art baselines regarding the explanation efficacy and efficiency. As for future directions, we consider extending CoRTX to multiple domains to obtain the general pre-trained latent explanation for downstream scenarios. It is also promising to explore healthcare applications with CoRTX framework, where the explanation labels require intensive human knowledge.

## 6 REPRODUCIBILITY STATEMENT

To ensure the reproducibility of our experiments and benefit the research community, we provide the source code of CoRTX along with the publicly available dataset. The detailed experiment settings, including hyper-parameters, baselines, and datasets, are documented in Appendix C to Appendix E. Our source code is accessible at: `https://github.com/ynchuang/CoRTX-720`.

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

APPENDIX

## A  PROOF OF THEOREM 1

We prove Theorem 1 in this section.

**Theorem 1** (**Compact Alignment**). *Let $f(\boldsymbol{x})$ be a $K_f$-Lipschitz [§] continuous function and $\boldsymbol{\Phi}(\boldsymbol{x}) = [\phi_1(\boldsymbol{x}), \cdots, \phi_M(\boldsymbol{x})]$ be the feature importance scores, where $\phi_i(\boldsymbol{x}) := \mathbb{E}_{\mathcal{S} \subseteq \mathcal{U} \backslash \{i\}} \big[ f(\widetilde{\boldsymbol{x}}_{\mathcal{S} \cup \{i\}}) - f(\widetilde{\boldsymbol{x}}_{\mathcal{S}}) \big]$. Given a perturbed positive instance $\widetilde{\boldsymbol{x}} \in \mathcal{X}^+$ satisfying $\min_{1 \leq i \leq M} \phi_i(\widetilde{\boldsymbol{x}}) \geq 0$, the explanation difference $\|\boldsymbol{\Phi}(\boldsymbol{x}) - \boldsymbol{\Phi}(\widetilde{\boldsymbol{x}})\|_2$ is bounded by the prediction difference $|f(\boldsymbol{x}) - f(\widetilde{\boldsymbol{x}})|$ as*

$$\|\boldsymbol{\Phi}(\boldsymbol{x}) - \boldsymbol{\Phi}(\widetilde{\boldsymbol{x}})\|_2 \leq (1 + \sqrt{2}\gamma_0)|f(\boldsymbol{x}) - f(\widetilde{\boldsymbol{x}})| + \sqrt{M}\gamma_0, \tag{7}$$

*where $\gamma_0 = K_f \|\boldsymbol{x}\|_2$ and $K_f \geq 0$ is the Lipschitz constant of function $f(\cdot)$.*

*Proof.* In order to calculate the importance score $\boldsymbol{\Phi}_i(\boldsymbol{x}) := \mathbb{E}_{\mathcal{S} \sim \mathcal{U} \backslash \{i\}} \big[ f(\widetilde{\boldsymbol{x}}_{\mathcal{S} \cup \{i\}}) - f(\widetilde{\boldsymbol{x}}_{\mathcal{S}}) \big]$ of the feature subset $\mathcal{S}$, we recast the formulation under the expression of $\mathbf{S} = \mathbf{1}_{\mathcal{S}} \in \{0, 1\}^M$ as follows:

$$\boldsymbol{\Phi}_i(\boldsymbol{x}) = \mathbb{E}_{\mathbf{S} \in \{0,1\}^{M-1}} \big[ f(\mathbf{S} \cup [1]_i \odot \boldsymbol{x} + (\mathbf{1} - \mathbf{S} \cup [1]_i) \odot \boldsymbol{x}_r) - f(\mathbf{S} \odot \boldsymbol{x} + (\mathbf{1} - \mathbf{S}) \odot \boldsymbol{x}_r) \big], \tag{8}$$

where $\mathbf{S} \cup [1]_i$ denotes not to mask feature $i$-th from perturbed masking $\mathbf{S}$.

Following Equation 8, we can now discuss explanation difference by each feature. Without lost of generality, we consider the explanation difference of a single feature $i$ as follows,

$$|\boldsymbol{\Phi}_i(\boldsymbol{x}) - \boldsymbol{\Phi}_i(\widetilde{\boldsymbol{x}})|$$
$$= \Big| \mathbb{E}_{\mathbf{S}} \big[ f(\mathbf{S} \cup [1]_i \odot \boldsymbol{x} + (\mathbf{1} - \mathbf{S} \cup [1]_i) \odot \boldsymbol{x}_r) - f(\mathbf{S} \odot \boldsymbol{x} + (\mathbf{1} - \mathbf{S}) \odot \boldsymbol{x}_r)$$
$$- \big( f(\mathbf{S} \cup [1]_i \odot \widetilde{\boldsymbol{x}} + (\mathbf{1} - \mathbf{S} \cup [1]_i) \odot \boldsymbol{x}_r) - f(\mathbf{S} \odot \widetilde{\boldsymbol{x}} + (\mathbf{1} - \mathbf{S}) \odot \boldsymbol{x}_r) \big) \big] \Big|$$
$$\leq \mathbb{E}_{\mathbf{S}} \Big[ \big| f(\mathbf{S} \cup [1]_i \odot \boldsymbol{x} + (\mathbf{1} - \mathbf{S} \cup [1]_i) \odot \boldsymbol{x}_r) - f(\mathbf{S} \odot \boldsymbol{x} + (\mathbf{1} - \mathbf{S}) \odot \boldsymbol{x}_r)$$
$$- \big( f(\mathbf{S} \cup [1]_i \odot \widetilde{\boldsymbol{x}} + (\mathbf{1} - \mathbf{S} \cup [1]_i) \odot \boldsymbol{x}_r) - f(\mathbf{S} \odot \widetilde{\boldsymbol{x}} + (\mathbf{1} - \mathbf{S}) \odot \boldsymbol{x}_r) \big) \big| \Big]$$
$$= \frac{1}{2^{M-1}} \Big[ \sum_{\mathbf{S}} |f(\mathbf{S} \cup [1]_i \odot \boldsymbol{x} + (\mathbf{1} - \mathbf{S} \cup [1]_i) \odot \boldsymbol{x}_r) - f(\mathbf{S} \cup [1]_i \odot \widetilde{\boldsymbol{x}} + (\mathbf{1} - \mathbf{S} \cup [1]_i) \odot \boldsymbol{x}_r)|$$
$$+ \sum_{\mathbf{S}} |f(\mathbf{S} \odot \widetilde{\boldsymbol{x}} + (\mathbf{1} - \mathbf{S}) \odot \boldsymbol{x}_r) - f(\mathbf{S} \odot \boldsymbol{x} + (\mathbf{1} - \mathbf{S}) \odot \boldsymbol{x}_r)| \Big] \tag{9}$$

Following Equation 9, the upper bound of the explanation difference can be derived by

$$|\boldsymbol{\Phi}_i(\boldsymbol{x}) - \boldsymbol{\Phi}_i(\widetilde{\boldsymbol{x}})|$$
$$\leq \frac{1}{2^{M-1}} \Big[ \sum_{\mathbf{S}} \big( |f(\mathbf{S} \cup [1]_i \odot \boldsymbol{x} + (\mathbf{1} - \mathbf{S} \cup [1]_i) \odot \boldsymbol{x}_r) - f(\mathbf{S} \cup [1]_i \odot \widetilde{\boldsymbol{x}} + (\mathbf{1} - \mathbf{S} \cup [1]_i) \odot \boldsymbol{x}_r)| \big)$$
$$+ \sum_{\mathbf{S}} K_f \| \mathbf{S} \odot \widetilde{\boldsymbol{x}} - \mathbf{S} \odot \boldsymbol{x} \|_2 \Big]$$
$$= \frac{1}{2^{M-1}} \Big[ \sum_{\mathbf{S}} |f(\mathbf{S} \cup [1]_i \odot \boldsymbol{x} + (\mathbf{1} - \mathbf{S} \cup [1]_i) \odot \boldsymbol{x}_r) - f(\mathbf{S} \cup [1]_i \odot \widetilde{\boldsymbol{x}} + (\mathbf{1} - \mathbf{S} \cup [1]_i) \odot \boldsymbol{x}_r)|$$
$$+ \sum_{\mathbf{S}} K_f \| \mathbf{S} \odot (\widetilde{\mathbf{S}} - \mathbf{1}) \odot \boldsymbol{x} \|_2 \Big]$$
$$\leq \underbrace{\frac{1}{2^{M-1}} \sum_{\mathbf{S}} \big( |f(\mathbf{S} \cup [1]_i \odot \boldsymbol{x} + (\mathbf{1} - \mathbf{S} \cup [1]_i) \odot \boldsymbol{x}_r) - f(\mathbf{S} \cup [1]_i \odot \widetilde{\boldsymbol{x}} + (\mathbf{1} - \mathbf{S} \cup [1]_i) \odot \boldsymbol{x}_r)| \big)}_{\delta_i}$$
$$+ K_f \|\boldsymbol{x}\|_2 \tag{10}$$

---

[§]Lipschitz continuity can be refered to (Virmaux & Scaman, 2018).

Note that the difference of prediction scores $|f(\boldsymbol{x}) - f(\widetilde{\boldsymbol{x}})|$ is equal to the summation of contribution score among all features (i.e., $|f(\boldsymbol{x}) - f(\widetilde{\boldsymbol{x}})| = |\sum_{i=1}^{M} \delta_i|$). Since we consider the assumption $\min_{1 \le i \le M} \boldsymbol{\Phi}_i(\widetilde{\boldsymbol{x}}) \ge 0$, we have $|\sum_{i=1}^{M} \delta_i| = \sum_{i=1}^{M} |\delta_i|$. In this manner, we have the upper bound of the explanation difference $\|\boldsymbol{\Phi}(\boldsymbol{x}) - \boldsymbol{\Phi}(\widetilde{\boldsymbol{x}})\|_2$ as follows:

$$
\begin{aligned}
\|\boldsymbol{\Phi}(\boldsymbol{x}) - \boldsymbol{\Phi}(\widetilde{\boldsymbol{x}})\|_2 &= \sqrt{\sum_{i=1}^{M} |\boldsymbol{\Phi}_i(\boldsymbol{x}) - \boldsymbol{\Phi}_i(\widetilde{\boldsymbol{x}})|^2} \\
&\le \sqrt{\sum_{i=1}^{M} (\delta_i + K_f \|\boldsymbol{x}\|_2)^2} \\
&\le \sqrt{\sum_{i=1}^{M} (\delta_i)^2 + \sqrt{2} \cdot (K_f \|\boldsymbol{x}\|_2) \cdot \sqrt{\sum_{i=1}^{M} \delta_i} + \sqrt{\sum_{i=1}^{M} (K_f \|\boldsymbol{x}\|_2)^2}} \\
&= \left\{ \|[\delta_1, \delta_2, \cdots, \delta_M]\|_2 + \sqrt{2} \gamma_0 |f(\boldsymbol{x}) - f(\widetilde{\boldsymbol{x}})| + \sqrt{M} \gamma_0 \right\} \\
&\le \left\{ \|[\delta_1, \delta_2, \cdots, \delta_M]\|_1 \right\} + \sqrt{2} \gamma_0 \cdot \left\{ |f(\boldsymbol{x}) - f(\widetilde{\boldsymbol{x}})| \right\} + \sqrt{M} \gamma_0 \\
&= (1 + \sqrt{2} \gamma_0) |f(\boldsymbol{x}) - f(\widetilde{\boldsymbol{x}})| + \sqrt{M} \gamma_0
\end{aligned}
$$

where $\gamma_0 = K_f \|\boldsymbol{x}\|_2$. $\qquad\square$

## B  PROOF OF THEOREM 2

**Theorem 2 (Explanation Error Bound).** *Let $\boldsymbol{\eta}(\cdot|\theta_{\boldsymbol{\eta}})$ be $K_{\eta}$-Lipschitz continuous, and $\hat{\boldsymbol{\Phi}}(\cdot) = \boldsymbol{\eta}(g(\cdot|\theta_g)|\theta_{\boldsymbol{\eta}})$ be the estimated explanations. If the encoder $g(\cdot|\theta_g)$ is well-trained, satisfying $\|\boldsymbol{\Phi}(\boldsymbol{x}) - \hat{\boldsymbol{\Phi}}(\boldsymbol{x})\|_2 \le \mathcal{E} \in \mathbb{R}^+$, then the explanation error on testing instance $\boldsymbol{x}_k$ can be bounded as:*

$$
\|\boldsymbol{\Phi}(\boldsymbol{x}_k) - \hat{\boldsymbol{\Phi}}(\boldsymbol{x}_k)\|_2 \le (1 + \sqrt{2} \gamma_k) |f(\boldsymbol{x}_k) - f(\widetilde{\boldsymbol{x}}_k^+)| + \sqrt{M} \gamma_k + \mathcal{E} + K_{\eta} \|\boldsymbol{h}_{\widetilde{\boldsymbol{x}}_k^+} - \boldsymbol{h}_{\boldsymbol{x}_k}\|_2, \quad (11)
$$

*where $\widetilde{\boldsymbol{x}}_k^+$ is a compact positive instance, $\boldsymbol{h}_{\boldsymbol{x}_k} = g(\boldsymbol{x}_k|\theta_g)$, and $\gamma_k = K_f \|\boldsymbol{x}_k\|_2$.*

*Proof.* Without loss of generality, we consider $\ell_2$ norm to evaluate the distance between predicted explanation scores and ground-truth explanation scores. For all $\boldsymbol{x}_k \in \mathcal{C}$, we have

$$
\begin{aligned}
\|\boldsymbol{\Phi}(\boldsymbol{x}_k) - \hat{\boldsymbol{\Phi}}(\boldsymbol{x}_k)\|_2 &= \|\boldsymbol{\Phi}(\boldsymbol{x}_k) - \boldsymbol{\Phi}(\widetilde{\boldsymbol{x}}_k^+) + \boldsymbol{\Phi}(\widetilde{\boldsymbol{x}}_k^+) - \hat{\boldsymbol{\Phi}}(\widetilde{\boldsymbol{x}}_k^+) + \hat{\boldsymbol{\Phi}}(\widetilde{\boldsymbol{x}}_k^+) - \hat{\boldsymbol{\Phi}}(\boldsymbol{x}_k)\|_2 \\
&\le \|\boldsymbol{\Phi}(\boldsymbol{x}_k) - \boldsymbol{\Phi}(\widetilde{\boldsymbol{x}}_k^+)\|_2 + \|\boldsymbol{\Phi}(\widetilde{\boldsymbol{x}}_k^+) - \hat{\boldsymbol{\Phi}}(\widetilde{\boldsymbol{x}}_k^+)\|_2 + \|\hat{\boldsymbol{\Phi}}(\widetilde{\boldsymbol{x}}_k^+) - \hat{\boldsymbol{\Phi}}(\boldsymbol{x}_k)\|_2 \\
&\le (1 + \sqrt{2} \gamma_k) |f(\boldsymbol{x}_k) - f(\widetilde{\boldsymbol{x}}_k^+)| + \sqrt{M} \gamma_k + \mathcal{E} + K_h \|g(\widetilde{\boldsymbol{x}}_k) - g(\boldsymbol{x}_k)\|_2 \\
&= (1 + \sqrt{2} \gamma_k) |f(\boldsymbol{x}_k) - f(\widetilde{\boldsymbol{x}}_k^+)| + \sqrt{M} \gamma_k + \mathcal{E} + K_h \|\boldsymbol{h}_{\widetilde{\boldsymbol{x}}_k} - \boldsymbol{h}_{\boldsymbol{x}_k}\|_2
\end{aligned}
$$

$\qquad\square$

## C  DETAILS ABOUT DATASETS

The experiments are conducted on two tabular datasets and one image dataset. The details of the datasets are provided as follows:

- **Census Income:** A collection of human social information with 26048 samples for training and validating; and 6513 samples for testing. Each instance has five continuous features and eight categorical features.

- **Bankruptcy:** A financial dataset contains 5455 samples of companies in the training set and validating set; and 1364 instances for the testing set. Each sample has 96 features characterizing each company and whether it went bankrupt.

- **CIFAR-10:** An image dataset with 60000 images in 10 different classes, where each image is composed of 32×32 pixels. We follow the original dataset division on training, validating, and testing process.

## D    DETAILS ABOUT BASELINE METHODS

Some of the baseline algorithms are implemented under the open-source package [¶] and the hyper-parameters are all decided with the optimal model convergence. In our experiments, the RTX frameworks, such as CoRTX and FastSHAP, are typically built on the training set and evaluated on the testing dataset.

**Tabular Dataset:** We here provide some detailed information about the baselines on tabular dataset. Supervised RTX: A supervised RTX-based MLP model trains with raw features of data instances and ground-truth explanation labels from scratch. The layer number in Supervised RTX is set to be six for purpose of fair comparison to CoRTX. FastSHAP: A state-of-the-art RTX method which adopts an DNN model to universally learn the distribution of approximated Shapley label (Jethani et al., 2021b). KernelSHAP (KS): The model proposes the weighted linear regressions to estimate the Shapley additive explanations from the prediction model scores (Kokhlikyan et al., 2020). Permutation Sampling (PS): PS estimates the feature attribution based on calculating the sensitivity gap of the scores from prediction model while inputting randomly masking data features (Mitchell et al., 2021). To achieve the competitive performance on the baselines, we demonstrate the following settings to the KS and PS in the tabular experiments. On Census Income dataset, the numbers of model evaluations on KS is set from $2^5$ to $2^{10}$, and the numbers of model evaluations on PS is given from $2^4$ to $2^{10}$. On Bankruptcy dataset, the s of model evaluations on KS is set from $2^3$ to $2^{11}$, and numbers of model evaluations on PS are from $2^3$ to $2^{10}$. For a fair comparison, CoRTX and Supervised RTX adopt the same loss function while conducting on the same explanation task.

**Image Dataset:** We evaluate the results with two non-amortized Shapley-based estimators: DeepSHAP and KS, where the first work utilizes a linear composition rule to estimate the Shapley values on the non-linear neural networks. Moreover, gradient-based methods are known for their efficiency compared to Shapley-based estimators. We list some gradient-based baselines: Saliency, Integrated Gradients (IG), SmoothGrad, and GradCAM. The mentioned four approaches derive the explanation based on the gradient values of prediction models. Lastly, to evaluate the efficacy of the image explainers, we adopt a state-of-the-art baseline, FastSHAP, to test the explanation generating speed and accuracy. To achieve the better performance on baselines, the evaluation times on baselines is basically decided on grid search once achieving competitive performance.

## E    EXPERIMENT DETAILS

We conduct the experiments on tabular dataset and image datasets. For the tabular dataset, we consider two common explanation tasks: feature attribution and importance ranking task. For the image dataset, the explanation results are presented using heatmap in the case study and evaluated by including and excluding the important pixels for perturbed image classification.

### E.1    EXPERIMENTS ON TABULAR DATASET

We verify the feature ranking task by using CoRTX-CE and the feature attribution task by exploiting CoRTX-MSE. The experiment on each tabular dataset follows the pipeline of ***Target Model Training***, ***Explanation Benchmarks*** and ***Explainer Implementing***. Each step is shown as follows.

***Target Model Training***: We exploit different prediction models $f(\cdot)$ on three datasets to evaluate the model-agnostic property of CoRTX. AutoInt (Song et al., 2019) is adopted for the Census Income dataset, and the MLP model is for the Bankruptcy dataset. The two prediction models are trained until the convergence, and the implementation is based on the DeepCTR [‖] package (Shen, 2017). Due to the tasks in two datasets, a binary cross entropy loss is given as the loss function of AutoInt and MLP model for the Census Income and Bankruptcy datasets, respectively. All hyper-parameters on target model training are decided by grid search throughout the classification results, including model layers, hidden units, etc.

***Explanation Benchmarks***: A brute-force algorithm for calculating the exact Shapley value is adopted for the Census Income dataset as the explanation labels. However, the Bankruptcy dataset contains

---

[¶] https://captum.ai
[‖] https://github.com/shenweichen/deepctr

many features (96 features), so it is hard to gain the exact Shapley value for the Bankruptcy dataset due to the extremely high computational cost. We hereby adopt the proximity of Shapley values as the explanation labels by using Antithetical Permutation Sampling(APS) (Mitchell et al., 2021; Lomeli et al., 2019) to convergence. This is because APS has shown to converge to exact Shapley values when countering high permutation times on tabular datasets. In this work, we set the model permutation times as $2 \times 10^5$ to generate the ground-truth labels.

***Explainer Implementation***: The explainer is implemented on the learning of latent explanation and fine-tuning. The explanation encoder and the explanation head are designed with the model structures and are learned with the hyper-parameters in Table 2. Following the existing works (Jethani et al., 2021b; Dhurandhar et al., 2018), we adjust the feature importance scores via additive efficient normalization (Ruiz et al., 1998) for the feature attribution task.

### E.2 EXPERIMENTS ON IMAGE DATASET

Our experiments on the image dataset are conducted under CoRTX-MSE. We focus on the feature attribution task on the image datasets, which follows the pipeline of ***Target Model Training***, ***Explanation Benchmarks*** and ***Explainer Implementing***. Each step is shown as follows.

***Target Model Training***: We utilize ResNet-18 as the prediction models $f(\cdot)$ in CIFAR-10 dataset. We train the target model from scratch until the convergence. All hyper-parameters on model training are decided by grid search throughout the classification results and follow the training setting of the target model from (Jethani et al., 2021b).

***Explanation Benchmarks***: We calculate the approximation to Shapley values as the training labels by adopting KS until the convergence (Covert & Lee, 2021; Jethani et al., 2021b). The estimated values from KS can infinitely approach exact Shapley values when encountering the optimal convergence (Lundberg & Lee, 2017). In this work, we set the model permutation times as 2048 to generate the ground-truth labels.

***Explainer Implementation***: The explanation encoder and the explanation head are designed with the model structures and are learned with the hyper-parameters in Table 2. Following the existing work (Jethani et al., 2021b), we adopt the surrogate model with the same structure as ResNet-18. The surrogate model is distilled from the original prediction model with masked input images to overcome the effect of out-of-distribution in masking perturbation (Frye et al., 2020). Following existing work (Jethani et al., 2021b), we adjust the feature attribution scores through additive efficient normalization (Ruiz et al., 1998) for generating the feature importance scores. Moreover, we follow the setting from the existing work (Jethani et al., 2021b) to conduct the bootstrapping evaluation. The total testing candidates are chosen to be 1000 and we randomly select 666 instances with 20 times repetitions for getting the bootstrap AUCs and Log-odds. Following the previous work in computer vision domain, the outcomes of CoRTX are post-processed through the moving average for smoothing visualization.

| | Dataset | Census Income | Bankrupcy | CIFAR-10 |
|---|---|---|---|---|
| | Target Model | AutoInt | MLP | ResNet-18 |
| | Explanation Encoder $g(\cdot)$ | 3-layer MLP | 6-layer MLP | ResNet-18 |
| | Optimizer | Adam | Adam | Adam |
| Explanation | Synthetic Positive Set | 30 | 300 | 300 |
| Representation | Batch Size | 1024 | 1024 | 1024 |
| Learning | Learning Rate | $5 \times 10^{-3}$ | $5 \times 10^{-3}$ | $5 \times 10^{-3}$ |
| | Temperature $\tau$ | 0.02 | 0.03 | 0.02 |
| | Binomial Distribution $\lambda$ | 0.5 | 0.5 | 0.5 |
| | Reference Value | Mean Value | Mean Value | Zero Value |
| | Explanation Head $\eta(\cdot)$ | 3-layer MLP | 3-layer MLP | 3-layer MLP |
| Fine-tuninig | Weight Decay$_{\text{CoRTX-MSE}}$ | $10^{-3}$ to $10^{-6}$ | $10^{-3}$ to $10^{-5}$ | $10^{-3}$ to $10^{-6}$ |
| | Weight Decay$_{\text{CoRTX-CE}}$ | 0 | 0 | – |

Table 2: Hyper-parameters and model structures settings in CoRTX.

### E.3 Computation Infrastructure

For a fair comparison of testing algorithmic throughput, the experiments are conducted based on the following physical computing infrastructure in Table 3.

| Device Attribute | Value |
|---|---|
| Computing infrastructure | GPU |
| GPU model | Nvidia-A40 |
| GPU number | 1 |
| GPU Memory | 46068 MB |

Table 3: Computing infrastructure for the experiments.

## F  Related Work

Machine learning has been widely applied in a variety of domains, such as recommender systems (Zha et al., 2022; Wang et al., 2020), anomaly detection (Li et al., 2021; Lai et al.), and voice recognition (Chandolikar et al., 2022; Minaee et al., 2023). Despite the advancements in ML, providing transparency in DNN models still remains a challenge, especially in yielding real-time model explanations for the deployment on real-time systems. Existing work aims to accelerate the derivation of explanations via two different ways: non-amortized methods and real-time explainers.

**Non-amortized Method.** Existing work of non-amortized methods can be categorized into three groups. The first group of methods adopts linear regressions to fit the non-amortized explanation, such as LIME (Ribeiro et al., 2016) and KS (Lundberg & Lee, 2017). Another group of methods adopts the preceding difference of the value function for the explanation, such as RISE (Petsiuk et al., 2018), Permutation Sampling (Mitchell et al., 2021) and SHEAR (Wang et al., 2022). The last group estimates the gradient towards the input data for the explanation, such as the GradCAM (Selvaraju et al., 2017), Integrated Gradient (Sundararajan et al., 2017) and SmoothGrad (Smilkov et al., 2017). Even though the non-amortized method can provide faithful explanation for DNN models, this group of methods suffers from high computational complexity since each data instance requires one non-amortized explainer to yield the explanation. Some gradient-based methods, such as (Sundararajan et al., 2017; Smilkov et al., 2017), are important explanation methods that provide a relatively faster explanation. However, gradient-based methods need extra time to perform the sampling process on each instance while generating the model explanation. The explanation derivation time is still highly dependent on the number of sampling and tested instances, which means that gradient-based methods are insufficient to be the real-time explainer.

**Real-time Explainer (RTX).** Unlike the non-amortized methods, the RTX framework maintains a unified explainer to generate the explanation among each data instance. This way, the explanation can be generated via a single feed-forward process of the explainer, which is much faster than the non-amortized methods but sightly degrades the efficacy. The learning strategy of the RTX framework formulates the explainer learning by giving strong assumptions on prior feature distribution (Chen et al., 2018; Dabkowski & Gal, 2017; Kanehira & Harada, 2019). Existing work (Chen et al., 2018) utilizes the instance-wise feature selection by maintaining a feature masking generator via maximizing the mutual information between selected features and corresponding labels. A well-trained feature masking generator is able to provide real-time explanation under a single feed-forward process. Another framework of RTX is to adopt the exact or approximated Shapley values as the ground-truth labels to learn the explainers (Wang et al., 2021; Jethani et al., 2021b; Covert et al., 2022; Schwab & Karlen, 2019). However, the exploitation of exact Shapley values suffers from extremely high computational complexity. To address this problem, FastSHAP (Jethani et al., 2021b) proposes a Monte-Carlo-based method to learn the explainer under the RTX framework. Specifically, FastSHAP generates the approximated Shapley values by randomly sampling batches of feature masks during the training process. Meanwhile, it updates the explainer to minimize the mean-square error between the overall contribution scores of masked features and outputs from the DNN explainer. FastSHAP enforces the explanation performance without utilizing the ground-truth Shapley value, which can be supported by the experiment results. The choice between non-amortized methods and RTX may depend on the use case (Rong et al., 2022b; Chuang et al., 2023), where there exists a trade-off between efficacy and efficiency on non-amortized methods and RTX.

# G    MEASUREMENT OF Log-odds

**Definition.** Log-odds indicates the confidence of the prediction given by the target model $f(\boldsymbol{x})$. Formally, Log-odds is defined as the log-likelihood ratio of $f(\boldsymbol{x})$ (Schwab & Karlen, 2019) as follows,

$$\text{Log-odds} = \log \frac{p}{1-p} \;, \tag{12}$$

where $p$ is the output probability of the target model $f(\boldsymbol{x})$.

**Inclusion and Exclusion AUC of Log-odds.** To evaluate the estimated feature attribution on image data, we conduct the exclusion and inclusion tests according to the feature importance scores. Once the important features are removed from the input instances, the target model $f(\boldsymbol{x})$ is expected to drop drastically on Log-odds which leads to a low exclusion AUC. On the contrary, it is expected to have a high inclusion AUC of Log-odds since the important features are orderly added back to the input instances of the target model. The remaining experimental settings follow FastSHAP (Jethani et al., 2021b).

**Inclusion and Exclusion AUC of $\Delta$Log-odds:** Besides exploiting Log-odds on measuring the performance, we additionally test CoRTX with $\Delta$**Log-odds** on the inclusion and exclusion task. $\Delta$**Log-odds** calculates the preceding difference between the Log-odds of original input instances and masked instances. Different from **Log-odds** and Top-1 Accuracy, Log-odds has the opposite indication where higher exclusion AUC and lower inclusion AUC represent better performance. Table 4 shows the results of the exclusion and inclusion under 1000 images. We observe that CoRTX outperforms other non-amortized baseline methods on both inclusion task and exclusion task. CoRTX performs better than FastSHAP on the inclusion task and is competitive with FastSHAP on the exclusion task.

|  | $\Delta$Log-odds | |
| --- | --- | --- |
|  | Exclusion | Inclusion |
| CoRTX | **2.790** $\pm$ 0.060 | **1.615** $\pm$ 0.026 |
| FastSHAP | **2.896** $\pm$ 0.045 | 1.642 $\pm$ 0.033 |
| KernelSHAP | 2.449 $\pm$ 0.055 | 1.687 $\pm$ 0.031 |
| Saliency | 2.075 $\pm$ 0.052 | 2.055 $\pm$ 0.036 |
| IG | 1.818 $\pm$ 0.053 | 2.040 $\pm$ 0.039 |
| Smoothgrad | 2.287 $\pm$ 0.054 | 2.111 $\pm$ 0.036 |
| Gradcam | 1.824 $\pm$ 0.049 | 2.062 $\pm$ 0.037 |
| Deepshap | 1.870 $\pm$ 0.054 | 2.040 $\pm$ 0.035 |

Table 4: Exclusion and Inclusion AUCs on Log-odds.

# H  ADDITIONAL RESULTS ON IMAGE DATASET

We demonstrate more explanation results on CIFAR-10 generated by CoRTX compared to other baselines. The results show that CoRTX can identify more human-understandable explanations toward other baselines.

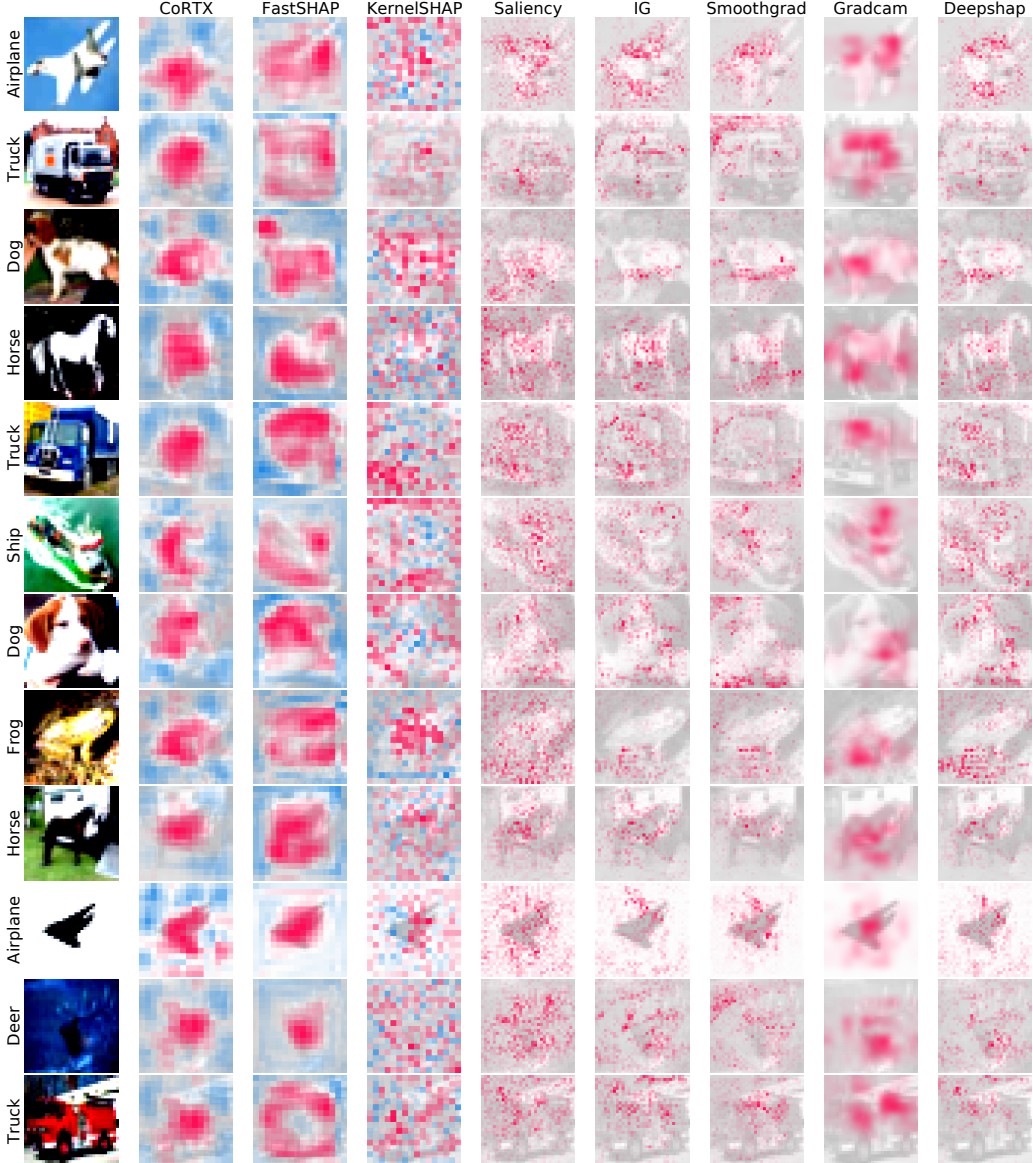

Figure 8: Explanations generated on CIFAR-10 Dataset.

# I ADDITIONAL BASELINE EXPERIMENTS

We conduct more experiments on CIFAR-10 dataset to compare CoRTX with two more baseline methods: RISE (Petsiuk et al., 2018) and XRAI (Kapishnikov et al., 2019). All hyper-parameters are decided with the optimal model convergence. The results show that CoRTX can provide more human-understandable explanations than RISE and XRAI.

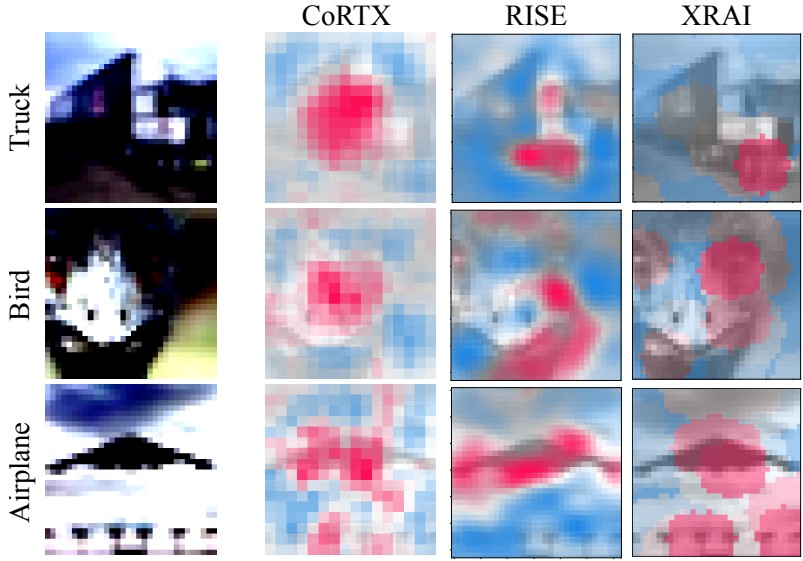

Figure 9: Comparison with the generated Explanation on CIFAR-10 Dataset.

We further conduct the experiments to compare L2X (Chen et al., 2018) with CoRTX on the Adult dataset. The hyper-parameters of L2X and CoRTX are all decided with the optimal model convergence. In the experiment, we choose Rank ACC and Faithfulness (Liu et al., 2021) as the metric, as the outputs from L2X are not Shapley-based values. Faithfulness is the metric to evaluate explanation tasks without any ground truth. As shown in Table 5, CoRTX outperforms L2X on two metrics, indicating the better capability for deriving model explanation.

| Methods | Rank ACC | Faithfulness |
|---|---|---|
| L2X (Chen et al., 2018) | $0.0847 \pm 0.0011$ | 0.542 |
| CoRTX-MSE | $0.6368 \pm 0.0067$ | **0.835** |
| CoRTX-CE | $\mathbf{0.7680} \pm 0.0031$ | – |

Table 5: Comparison with L2X on Adult dataset.

## J  EVALUATION OF CORTX ON SURROGATE MODELS

Besides the quantitative evaluation setting in Section 4.4.2 that follows (Rong et al., 2022a). We further conduct experiments by retraining the surrogate model, where the experiment settings follow from (Jethani et al., 2021b;a). The following table shows the results of the exclusion and inclusion AUC under 1000 images. We observe that CoRTX outperforms other non-amortized baselines on both inclusion and exclusion tasks. CoRTX performs better than FastSHAP (a SOTA of supervised RTX method) on the exclusion task and is competitive with FastSHAP on the inclusion task. The results reveal a similar pattern as using the original prediction model in the quantitative experiments, which proves that CoRTX has the capability to derive effective model explanations on either of the experiment settings.

| | Top-1 Accuracy | |
| --- | --- | --- |
| | Exclusion | Inclusion |
| CoRTX | $0.372 \pm 0.012$ | $0.772 \pm 0.012$ |
| FastSHAP | $0.386 \pm 0.012$ | $0.767 \pm 0.013$ |
| KernelSHAP | $0.406 \pm 0.011$ | $0.767 \pm 0.012$ |
| Saliency | $0.543 \pm 0.012$ | $0.703 \pm 0.012$ |
| IG | $0.581 \pm 0.012$ | $0.704 \pm 0.012$ |
| Smoothgrad | $0.480 \pm 0.012$ | $0.695 \pm 0.012$ |
| Gradcam | $0.573 \pm 0.012$ | $0.719 \pm 0.013$ |
| Deepshap | $0.573 \pm 0.012$ | $0.706 \pm 0.012$ |

Table 6: Evaluation of CoRTX on Surrogate Models.

## K  $\ell_2$-ERROR OF PS/KS WITH HIGH PERMUTATION

We conducted additional experiments on the Adult dataset to show that PS/KS with high permutation is very similar to the exact values of Shapley values, especially when the perturbation is above 4000 times. This shows that high perturbated KS/PS is sufficient to generate the ground truth. There is no need to do higher permutation sampling for KS/PS to generate the ground truth. This is because the error remains extremely small as well, but the execution time grows drastically.

| Number of permutation | 2000 | 4000 | 6000 | 8000 |
| --- | --- | --- | --- | --- |
| KernelSHAP ($\ell_2$-error) | 0.0038 | 0.0027 | 0.0021 | 0.0018 |
| APS ($\ell_2$-error) | 0.00031 | 0.00025 | 0.00020 | 0.00017 |
| KernelSHAP (sec/instance) | 0.3351 | 0.6816 | 0.9869 | 1.3089 |
| APS (sec/instance) | 0.0621 | 0.1164 | 0.1705 | 0.2222 |

Table 7: $\ell_2$-error of PS/KS with High Permutation.

## L  ADDITIONAL NOTATION CLARIFICATION

In this section, Table 8 shows the detailed information of the notations in Section 4.2 for clarification.

| Notation | Definition & Description | Dimension |
| --- | --- | --- |
| $\hat{\mathbf{r}}_j$ | Predicted ranking index score list of feature $x_j$ from CoRTX. Each index score in $\hat{\mathbf{r}}_j$ represents the score of feature $x_j$ that is ranked at this index. | $\hat{\mathbf{r}}_j \in \mathbb{R}^M$ |
| $\hat{\mathrm{r}}_j$ | Predicted ranking index from CoRTX, where $\hat{\mathrm{r}}_j = \arg\max \hat{\mathbf{r}}_j$ for $1 \leq j \leq M$ | $\hat{\mathrm{r}}_j \in \mathbb{R}^M$ |
| $\mathrm{r}_j$ | The ground-truth ranking label sorted from Shapley values. | $\mathrm{r}_j \in \mathbb{R}^M$ |

Table 8: Notations in Section 4.2.

# M $\ell_2$-ERROR AND EXECUTION TIME OF PS/KS WITH HIGH PERMUTATION

We conduct an experiment on the Adult dataset and indicate the explanation performance and the execution time of PS/KS and CoRTX in the following table. It is observed that there is a trade-off between the speed and performance of KS/PS. A fair comparison should keep the execution time as close as possible. According to the results, CoRTX generates the model explanation in around 0.00015 seconds per instance, while high-permutated PS/KS requires at least 2000x longer execution time. We believe that comparing CoRTX with low-permutated KS/PS is fairer than with high-permutated KS/PS. All experimental settings follow FastSHAP (Jethani et al., 2021b).

|  | KS-300 | KS-2000 | KS-4000 | KS-6000 | KS-8000 | CoRTX-MSE |
|---|---|---|---|---|---|---|
| $\ell_2$-error | 0.0189 | 0.0038 | 0.0027 | 0.0021 | 0.0018 | 0.007 |
| Time (sec) | 0.1477 | 0.3351 | 0.6816 | 0.9869 | 1.3089 | 0.00015 |

|  | APS-300 | APS-2000 | APS-4000 | APS-6000 | APS-8000 | CoRTX-MSE |
|---|---|---|---|---|---|---|
| $\ell_2$-error | 0.0171 | 0.00031 | 0.00025 | 0.00020 | 0.00017 | 0.007 |
| Time (sec) | 0.0380 | 0.0621 | 0.1164 | 0.1705 | 0.2222 | 0.00015 |

Table 9: $\ell_2$-error and execution time of PS/KS with high permutation.

