# OpenReview forum: "CoRTX: Contrastive Framework for Real-time Explanation"
_ICLR.cc/2023/Conference — ICLR 2023 poster_

### Official Review · Reviewer_ZCbd · 2022-10-22

**Confidence:** 3
**Clarity, Quality, Novelty And Reproducibility:** Have met the standards.
**Correctness:** 3
**Technical Novelty And Significance:** 3
**Empirical Novelty And Significance:** 3
**Recommendation:** 8

**Strength And Weaknesses:**

- This paper is well-written, and the structure is clear. The authors extend from the problem in explainable machine learning of the huge requirement of labels to why unsupervised contrastive learning can help explanation with clear theoretical analysis.
- Settings and evaluation metrics in experiments are good. The formulated three questions do help understand the motivation and effectiveness of this method, including the efficiency and efficacy of CoRTX, and whether the extracted contrastive explanation representation is still effective when fine-tuning, also the ablations on synthetic positive augmentations are good.
- The generality of the proposed approach is well demonstrated by considering experimental scenarios with various modalities, i.e., experiments on images and tabular data.

Cons:

- With the mention of "real-time" and "in practice" in the Intro, this paper focus on the real-time scenario, which will easily remind the readers of industrial-level real-time explanations, e.g., plug-and-play model or tool. However, if one involves a training phase, so-called "real-time" is not achievable, even though only the explanation head is trained in CoRTX. Although the proposed components of light-label manner and synthetic positive augmentation do reduce the computational effort and prevent the instances from being too hard in contrastive learning, it is still not the same as what I perceive as real-time for real-world scenarios. Besides, the evaluated datasets are not "in practice". To me, this is a good paper on explainable machine learning, but not a paper on "real-time". I would recommend that the authors change this paper's stance and contribution in another way rather than "real-time".
- Sections 2.2 and 2.3 depict the motivation of the whole method, i.e., combining the advantages of two conventional approaches and reducing the dependency on labels. Although the motivation is intuitive, the content is not convincing enough. I would recommend the authors could write in a better way to couple unsupervised contrastive learning with explainable ML. For example, labels may bring strong supervised information into the training phase, affecting the inductive bias of the model, making the model favor more readily available label information over the information that may help in explanation, where the contrastive learning has already proved that models can learn features that are beneficial for downstream tasks without relying on labels.
- The experiments and models are well designed, but there may be a lot of room for tuning in terms of parameters and design, such as the choice of the explanation head. The choice of the number of fully connected layers may have an impact, as also the dimensions of these layers, e.g., multiple projection heads of SimCLR.


**Summary Of The Paper:**

This paper proposes Constrastive Real-time Explanation (CoRTX) framework to avoid encountering efficiency issues in practical scenarios. Specifically, to avoid the requirement of large amounts of explanation labels, this paper designs a synthetic strategy to select positive and negative samples for contrastive explanation representation learning. With experiments on three datasets, and theoretical analysis on both the positive sample selection method and contrastive explanation error bound, the proposed method CoRTX demonstrates efficiency and efficacy.

**Summary Of The Review:**

Overall, I think the studied problem is essential, and the proposed method is novel and reasonable. The motivations are good, and the experiments are clear, comprehensive, and convincing. The theoretical analysis also helps much in understanding how contrastive learning help in explanation. Though the claims in motivations are not entirely convincing, I believe this is a good paper.

---

> ### Author Response · Authors · 2022-11-13
> **Response to Reviewer ZCbd.**
>
> We thank the reviewer for the constructive comments and appreciate the reviewer for the recognition of the effectiveness of our work.
>
> **Q1: With the mention of "real-time" and "in practice" in the Intro, this paper focus on the real-time scenario, which will easily remind the readers of industrial-level real-time explanations, e.g., plug-and-play model or tool. However, if one involves a training phase, so-called "real-time" is not achievable, even though only the explanation head is trained in CoRTX. I would recommend that the authors change this paper's stance and contribution in another way rather than "real-time".**
>
> [A1]: Thanks for pointing this out. We would like to clarify that a real-time system typically refers to the system obtaining real-time inference processes to provide the service. This is because the prediction model in a real-time system is usually an offline pre-trained model. There is no need to train the prediction model during the inference stage. Generally, traditional local explanation methods[1,2,3] generate the explanation instance by instance. Each tested instance is required to learn its unique explainer during the explanation derivating time. This makes the local ones the non-real-time explainers. On the contrary, CoRTX can provide multiple explanations of a batch of instances simultaneously, and it does not need to train the explainer during the explanation-generating time. In this manner, CoRTX is a real-time explainer since it completes the explanation inference in a very short time.
>
> | Method | IG | SmoothGrad | CoRTX |
> | :----: | :----: | :----: | :----: |
> | Time (**ms per image**) | 54.6 | 60.0 | 0.4 |
>
> [1] Lundberg, Scott M., and Su-In Lee. "A unified approach to interpreting model predictions." Advances in neural information processing systems 30 (2017).
>
> [2] Mitchell, Rory, et al. "Sampling permutations for Shapley value estimation." (2022): 1-46.
>
> [3] Selvaraju, Ramprasaath R., et al. "Grad-cam: Visual explanations from deep networks via gradient-based localization." Proceedings of the IEEE international conference on computer vision. 2017.
>
> **Q2: I would recommend the authors could write in a better way to couple unsupervised contrastive learning with explainable ML.**
>
> [A2]: Thanks for the constructive suggestions. We have revised the related paragraph according to the comments and the revised sentences are highlighted in blue in Section 2.2.
>
> **Q3: The experiments and models are well designed, but there may be a lot of room for tuning in terms of parameters and design, such as the choice of the explanation head. The choice of the number of fully connected layers may have an impact, as also the dimensions of these layers, e.g., multiple projection heads of SimCLR.**
>
> [A3]: Thanks for the comments. We agree with the statement from the reviewer. We have conducted experiments to show the sensitivity check on layer number in the explanation head. We test CoRTX-CE with 1-layer, 3-layer, and 5-layer explanation head on the Adult dataset. Due to the time limitation of rebuttal, the experiment is conducted on feature importance ranking tasks. The results are shown in the following table, which shows 3-layer explanation head provides better performance than the 1-layer one and 5-layer one. The additional experiment results are concluded in Appendix I. We thank you again for your comments.
>
> | CoRTX-CE | 1-layer | 3-layer | 5-layer |
> | :----: | :----: | :----: | :----: |
> | Rank ACC | 0.6893 $\pm$ 0.0024 |  0.7690 $\pm$ 0.0028 | 0.7628 $\pm$ 0.0029 |

---

> ### Author Response · Authors · 2022-11-18
> **To reviewer ZCbd**
>
> Dear reviewer ZCbd,
>
> Thanks again for your valued comments on our work. We have responded to your initial comments. As the deadline for context revising (Nov. 18th) is coming, we are looking forward to your feedback and will be happy to answer any further questions you may have.
>
> Sincerely,
>
> Authors

---

> > ### Comment · Reviewer_ZCbd · 2022-11-21
> > **post-rebuttal reply**
> >
> > Happy to keep my score. Best of luck!

---

### Official Review · Reviewer_bBUe · 2022-10-24

**Confidence:** 4
**Correctness:** 2
**Technical Novelty And Significance:** 2
**Empirical Novelty And Significance:** 2
**Recommendation:** 6

**Clarity, Quality, Novelty And Reproducibility:**

Clarity:
The paper is not very clear.
Section 3 is too brief in explaining the architecture of the framework, especially when discussing the positive examples.
x+ is never clearly stated how it is produced, and the reader must infer this from Figure 2.
The parameters of the equation are sometimes omitted, such as M and r in equation 2.
Finally, the figures are not very clear.
In Figure 1, it would be nice to see the execution time to understand whether supervised approaches are really limited in a real-time context.
In Figure 3 it is not clear what a vertical line means, does it have infinite throughput? It is not clear.
Quality: the quality of the paper could be improved.
Novelty:
The work is not too new as it combines existing approaches to improve the throughput of their algorithm.
Reproducibility:
The code and all the information to reproduce the results is provided.

**Strength And Weaknesses:**

Strengths:
+ The results are very good and CoRTX outperforms the tested methods. The evaluations were performed correctly, including the evaluation of errors.
+ Quantitative evaluation of the feature importance map is well conducted according to current standards.
+ Good number of data sets tested to validate the claims.

- The fine tuning part is done according to the tested metric. This is done only for CoRTX and is not fair to compare against other methods tested.
- Two types of approaches were described in the introduction: methods that learn an explainer based on labels and supervised methods. CoRTX is a semi-supervised method, but only the first type of methods is tested. I would like to see a comparison with methods such as Dabkowski & Gal 2017, Chen et al. 2018 and Kanehira & Harada 2019, presented in the introduction and never mentioned again. CoRTX uses a similar framework as Chen et al. but the comparison was not made.
- The saliency maps produced (Figure 5) are very different from those compared. I suggest a comparison with masking methods such as RISE (https://es.sonicurlprotection-fra.com/click?PV=2&MSGID=202210211531580778554&URLID=35&ESV=10.0.18.7423&IV=C019EBF33BCE13D9800D210395403F1C&TT=1666366319343&ESN=UynqeUYHEpb%2Ba0WBCa0%2FCeGBiM%2Fl9mITsoFXluKYkDY%3D&KV=1536961729280&B64_ENCODED_URL=aHR0cHM6Ly9naXRodWIuY29tL2VjbGlxdWUvUklTRQ&HK=7F3046409B02770A56F86E7E471588356C1EC6136067365613FD327EB358B71D) or XRAI (https://es.sonicurlprotection-fra.com/click?PV=2&MSGID=202210211531580778554&URLID=34&ESV=10.0.18.7423&IV=CC9762D41E2E67C8D730D7F2EE59A2AE&TT=1666366319343&ESN=kNkzfYKleQHNaDAOrggEOdhqRAXo1UQxQGUwd7mAT7g%3D&KV=1536961729280&B64_ENCODED_URL=aHR0cHM6Ly9naXRodWIuY29tL2h1bW1hdC9zYWxpZW5jeQ&HK=225CD5F9DF0A3A0AAC78077D46BE0581FB3486DED2BEC28FDA05130432137D5B) that produce saliency maps very similar to those of CoRTX.
- The qualitative assessment of saliency maps has not been adequately evaluated. In particular, Section 4.4.1 states that "CoRTX provides better explanation results and highlights particular regions that are essential for model predictions." However, shifting salience toward the subject is not always a symptom of better explanations. There is no certainty that the subject is important for prediction. In particular, there is an article entitled "Sanity check for saliency maps" by Adebayo et al. that shows this detrimental behavior of methods such as Smoothgrad (tested in this article).
- The fine-tuning part appears to be optional from the text, but instead is required to produce a saliency map of size M. The representations output by the encoder have size d and cannot be used as saliency maps because d << M.

**Summary Of The Paper:**

This paper presents a new real-time explainer framework called COntrastive Real-Time eXplanation (CoRTX).
The main goal of the algorithm is to limit the dependence of real-time explainers on predefined labels and to improve the throughput (number of samples/time) of the algorithm.
CoRTX designs a positive and negative synthetic sampling strategy to learn an explanation representation in the form of feature importance.
A positive sample is a masking of the original data to explain where some features are replaced by zero values.
This explanation representation is shown to be an approximation of Shapley values with a set of theorems.
The architecture of the method is very similar to that of an autoencoder, but with only the encoder and a loss of similarity.
Learning is performed encouraging representations of positive samples to be close, while those of negative samples to be far away.
After learning the representations, a fine-tuning process with a limited amount of ground truth labels is carried out to refine the explanations.

**Summary Of The Review:**

The paper is not very clear and needs a thorough rewrite; the experiments part needs further experiments to fully support the claims.

---

> ### Author Response · Authors · 2022-11-13
> **Response to Reviewer bBUe. [Part 1/3 Q1 Q2]**
>
> We thank the reviewer for the constructive comments.
>
> **Q1: The fine tuning part is done according to the tested metric. This is done only for CoRTX and is not fair to compare against other methods tested.**
>
> [A1]: Thanks for the comment. We believe the comparison is fair based on task level and method level reasons.
>
>   - **Task level:** Most explanation tasks in prior arts focus on the Feature Attribution Task (e.g., FastSHAP[1] and KernelSHAP[2]), where the existing explanation methods are designed according to the Feature Attribution Task. In our work, CoRTX-MSE yields feature attribution scores according to the given instances, which share the same design philosophy of existing methods[1,2]. The attribution scores of CoRTX-MSE can be directedly adopted in Feature Attribution Task and Feature Importance Ranking Task. Thus, the comparison between CoRTX-MSE and other baselines developed according to the attribution task is fair. We further propose CoRTX-CE to provide a better solution for the Feature Importance Ranking Task since some prior achievements focus on the importance ranking task[3].
>
>   - **Method level:** CoRTX shares a similar training philosophy with other supervised paradigms of RTX but only a difference in the learning process. CoRTX and other supervised paradigms of RTX are adjusted by the explanation labels during the training phase. CoRTX is a two-step learning explainer, which first learns the representation of the explanation and then uses a few labels to adjust the explainer for model explanation generation. Other supervised paradigms of RTX (e.g., FastSHAP and L2X) are typically end-to-end learning models that calibrate the explainer based on all labels.
>
> [1] Lundberg, Scott M., and Su-In Lee. "A unified approach to interpreting model predictions." Advances in neural information processing systems 30 (2017).
>
> [2] Jethani, Neil, et al. "FastSHAP: Real-Time Shapley Value Estimation." International Conference on Learning Representations. 2021.
>
> [3] Wang, Guanchu, et al. "Accelerating Shapley Explanation via Contributive Cooperator Selection." International Conference on Machine Learning. PMLR, 2022.
>
>
> **Q2: Two types of approaches were described in the introduction: methods that learn an explainer based on labels and supervised methods. CoRTX is a semi-supervised method, but only the first type of methods is tested. I would like to see a comparison with methods such as Dabkowski & Gal 2017, Chen et al. 2018 and Kanehira & Harada 2019, presented in the introduction and never mentioned again. CoRTX uses a similar framework as Chen et al. but the comparison was not made.**
>
> [A2]: Thanks for the constructive feedback.
> We would like to clarify that L2X is different from CoRTX. CoRTX provides an explanation encoder to learn the explanation representation, and then we use the pre-trained explanation representation to fine-tune with very few explanation labels by explanation head. Explanation representation in CoRTX is a latent vector projecting the explanation information to low-rank space. In contrast, L2x directly learns the feature selection model to yield the importance scores of each feature, which process the information on the original feature space.
>
> We conduct additional experiments to compare CoRTX with L2X[1], categorized as feature-selection-based approaches on the Adult dataset. The following table reveals the results of the new experiments. The results show that learning the explanation information in a low-rank space is better than in the original feature space.
>
> | Methods | Rank ACC | Faithfulness[2] |
> | :----: | :----: | :----: |
> | L2X | 0.0847 $\pm$ 0.0011 | 0.54 |
> | CoRTX-MSE | 0.6368 $\pm$ 0.0067 | **0.83** |
> | CoRTX-CE | **0.7680 $\pm$ 0.0031** | ----- |
>
> [1] Chen, Jianbo, et al. "Learning to explain: An information-theoretic perspective on model interpretation." International Conference on Machine Learning. PMLR, 2018.
>
> [2] Liu, Yang, et al. "Synthetic Benchmarks for Scientific Research in Explainable Machine Learning." Thirty-fifth Conference on Neural Information Processing Systems Datasets and Benchmarks Track (Round 2). 2021.

---

> ### Author Response · Authors · 2022-11-13
> **Response to Reviewer bBUe. [Part 2/3 Q3 Q4 Q5]**
>
> **Q3: The qualitative assessment of saliency maps has not been adequately evaluated. There is no certainty that the subject is important for prediction.**
>
> [A3]: Thanks for the constructive comments. We agree that qualitative assessment is not the best evaluation way to model explanations [1]. We provide the case study following the SoTA methods[2,3,4] and the results are just for reference and comparison to the other baselines. Instead, we provide the quantitative results on the image dataset by following the experiment settings from [1,3]. The quantitative results show that CoRTX is competitive to FastSHAP[1] and outperforms the rest of the baselines. We have revised the words mentioned in Section 4.4.1 and the revised sentences (i.e., CoRTX provides better human-understandable explanation results.) are highlighted in blue.
>
> [1] Adebayo, Julius, et al. "Sanity checks for saliency maps." Advances in neural information processing systems 31 (2018).
>
> [2] Lundberg, Scott M., and Su-In Lee. "A unified approach to interpreting model predictions." Advances in neural information processing systems 30 (2017).
>
> [3] Jethani, Neil, et al. "FastSHAP: Real-Time Shapley Value Estimation." International Conference on Learning Representations. 2021.
>
> [4] Petsiuk, Vitali, Abir Das, and Kate Saenko. "Rise: Randomized input sampling for explanation of black-box models." arXiv preprint arXiv:1806.07421 (2018).
>
> **Q4: The fine-tuning part appears to be optional from the text, but instead is required to produce a saliency map of size M. The representations output by the encoder have size d and cannot be used as saliency maps because d << M.**
>
> [A4]: Thanks for pointing out this question. We would like to clarify that fine-tuning part is required but not optional. CoRTX does not directly use explanation representation as the predicted model explanation. Instead, the explanation representation is further fine-tuned by the explanation head, and then the explanation head generates the final model explanation. We believe that the reviewer misunderstood our proposed CoRTX framework due to Figure 2. Thus, we have revised Figure 2 in case of further misreading information.
>
> **Q5: Some notations and concepts clarification.**
>
> - **Section 3 is too brief in explaining the architecture of the framework, especially when discussing the positive examples.**
>
>   [A5-1]: This part is revised according to the suggestions and highlighted in blue in Section 3.
>
> - **x+ is never clearly stated how it is produced, and the reader must infer this from Figure 2.**
>
>   [A5-2]: Thanks for pointing this out. We would like to clarify that there is no notation “x+” in our context. We guess that the reviewer wants to know more about the definition of either the set of synthetic positive instances $\mathcal{X}^{+}$ or the positive instances $\widetilde{\bf{x}}^{+}$. The definition of $\mathcal{X}^{+}$ is stated in Equation (2) and the selection strategy of $\widetilde{\bf{x}}^{+}$ is provided in Equation (4).
>
> - **The parameters of the equation are sometimes omitted, such as M and r in equation 2.**
>
>   [A5-3]: Thanks for pointing this out. We would like to clarify that $M$ and $r$ are stated in Section 2.1. $M$ represents the feature number of each instance and $r$ is used for denoting the reference values $\boldsymbol{x}_r$, where $\boldsymbol{x}_r = \mathbb{E}[ \boldsymbol{x} \mid \boldsymbol{x} \sim P(\boldsymbol{x})]$ denotes the reference values from feature distribution $P(\boldsymbol{x})$.
>
> - **In Figure 1, it would be nice to see the execution time to understand whether supervised approaches are really limited in a real-time context.**
>
>   [A5-4]: Thanks for pointing this out. We would like to clarify the misunderstanding from the reviewer on supervised RTX. Supervised RTX is applicable to derive real-time explanation via the single feed-forward process, which obtains a similar inference time to CoRTX. The limitation of supervised RTX is not the real-time context but the requirement of large quantities of label usage during the training stage.
>
> - **In Figure 3 it is not clear what a vertical line means, does it have infinite throughput.**
>
>   [A5-5]: Thanks for pointing this out. The vertical line is only the direction line for marking the exact values of the given metrics. We have revised Figure 3 to prevent the misunderstanding.

---

> ### Author Response · Authors · 2022-11-13
> **Response to Reviewer bBUe. [Part 3/3 Q6 Q7]**
>
> **Q6: Novelty: The work is not too new as it combines existing approaches to improve the throughput of their algorithm.**
>
> [A6]: We respectfully disagree with the statement from the reviewer. CoRTX is not a naive combination of existing algorithms. The contributions of CoRTX are as follows:
>
>   - Motivated by the difficulties of gaining explanation labels, CoRTX provides a contrastive framework for deriving explanation representation, which can effectively reduce the required amounts of explanation labels. Traditional contrastive learning framework cannot directly apply to XAI tasks since the representation generated from traditional ones obtains no explanation information.
>
>   - The explanation-oriented data augmentation strategy is proposed to select the positive pair for the contrastive learning paradigm in CoRTX, which benefits encoding the explanation information into the learned representation. The learned representation is helpful for deriving model explanations during the fine-tuning phase with very few amounts of explanation labels.
>
>   - Theoretical analysis indicates that CoRTX can effectively learn the explanation representation over the training set and strictly bound the explanation error.
>
> To sum up, we believe that our proposed CoRTX is not a trivial combination of the existing works. Motivated by contrastive learning, we develop a new contrastive framework for deriving the model explanation in a real-time estimation. The theoretical statements support the efficacy and efficiency of our proposed CoRTX.
>
> **Q7: Additional comparison and adjustment for experiments.**
>
> - **The saliency maps produced (Figure 5) are very different from those compared. I suggest a comparison with masking methods such as RISE or XRAI.**
>
>   [A7]: Thanks for the comments. We are unsure of the "difference" mentioned by the reviewer since our baseline results in Section 4.4.1 are primarily consistent with the results from FastSHAP[1]. We believe that only the results from GradCAM in our work are different from FastSHAP since we did not previously adopt any computer vision post-processing techniques. To prevent misunderstanding, we have updated the new results by applying the same post-processing techniques of FastSHAP and highlighted them in blue for recognition. If the difference you mentioned is from the original paper, we believe that one of the possible reasons is the resolution of the tested instances. High-resolution images are able to provide better-visualizing effects; however, low-resolution images obtain poor visualization capability. We have conducted two more experiments compared with the baselines mentioned by the reviewer. Please refer to the saliency maps results on RISE[2] and XRAI[3] in Appendix $\mathrm{I}$. We also updated the additional experiments in Appendix $\mathrm{I}$ for reference.
>
> [1] Jethani, Neil, et al. "FastSHAP: Real-Time Shapley Value Estimation." International Conference on Learning Representations. 2021.
>
> [2] Petsiuk, Vitali, Abir Das, and Kate Saenko. "Rise: Randomized input sampling for explanation of black-box models." arXiv preprint arXiv:1806.07421 (2018).
>
> [3] Kapishnikov, Andrei, et al. "Xrai: Better attributions through regions." Proceedings of the IEEE/CVF International Conference on Computer Vision. 2019.

---

> ### Author Response · Authors · 2022-11-18
> **To reviewer bBUe**
>
> Dear reviewer bBUe,
>
> Thanks again for your valued comments on our work. We have responded to your initial comments. As the deadline for context revising (Nov. 18th) is coming, we are looking forward to your feedback and will be happy to answer any further questions you may have.
>
> Sincerely,
>
> Authors

---

> > ### Comment · Reviewer_bBUe · 2022-11-23
> > **On the responses**
> >
> > I would like to thank the authors for their very responses to all the points raised. These adjustments increased the evaluation of the paper.

---

### Official Review · Reviewer_A6qm · 2022-10-25

**Confidence:** 3
**Correctness:** 3
**Technical Novelty And Significance:** 3
**Empirical Novelty And Significance:** Not applicable
**Recommendation:** 6

**Clarity, Quality, Novelty And Reproducibility:**

Clarity. The paper is well organized.
Quality. The paper appears technically sound, but I have not carefully checked the maths/details.
Novelty. The paper contributes some new ideas.
Reproducibility. Good: key resources (e.g., proofs, code, data) are available, and key details (e.g., proofs, experimental setup) are sufficiently well-described for competent researchers to reproduce the main results confidently.

**Strength And Weaknesses:**

Strength
The paper focused on an important problem.
The synthetic strategy to select positive instances for contrastive learning is reasonable and well backed up, theoretically and experimentally.
The experiments are well established to show the effectiveness of the proposed method.

Weakness
The related works are not sufficient compared to the scope of the title and abstract. In particular, the authors mainly focused on such Shapley-value and aimed to predict these values in test time. However, many other approaches (e.g., gradient-based saliency methods) can also provide fast real-time explanations. Some discussions are needed.
I would prefer more intuition/explanation of how contrastive learning, especially the instances selection strategy, is useful in learning effective representation for the explainer.
In the abstract, the authors claimed that “accurate explanation labels are hard to obtain due to limited human efforts.” Is this true? Since the explanation is of the black-box model, I do not see how humans take part in producing the explanation labels.
In Figure 5, it would be useful to include the result of SHAP so that the readers can evaluate how the proposed method’s explanation compares to the ground-truth explanation.
In the Feature Importance Ranking Task (Section 4.2), the equation to obtain hat{r}j is not clear (although I can guess what the authors are going to do).

**Summary Of The Paper:**

The authors proposed a method to obtain a good and fast explanation at the testing time. In particular, the authors focused on the approach that learns an explainer model to mimic the “accurate explanation” at test time (e.g., predict the Shapley value).
The authors pointed out that a major issue of existing methods is the requirement of many ground-truth explanation labels (i.e., fully accurate Shapley value for each sample), which can be computationally expensive.
The authors proposed a contrastive learning method to learn useful explanation-oriented representations so that they can learn the explainer effectively with only a few explanation labels. The main contribution is the synthetic strategy to select positive and negative instances.
The authors conducted various experiments to verify the efficiency of the proposed method.

**Summary Of The Review:**

The authors focused on an important problem of acquiring fast explanations at test time, although the scope is limited to Shapley-liked explanations. The proposed method is yet simple but effective and well backed up theoretically and experimentally. The weaknesses seem fixable, and under the condition that these points improve, I will recommend acceptance.

---

> ### Author Response · Authors · 2022-11-13
> **Response to Reviewer A6qm. [Part 1/2 Q1 Q2]**
>
> We thank the reviewer for the constructive comments and appreciate the reviewer for the recognition of the effectiveness and efficiency of our work.
>
> **Q1: The related works are not sufficient compared to the scope of the title and abstract. (e.g., gradient-based saliency methods).**
>
> [A1]: Thanks for your comment. We agree that gradient-based saliency methods are important explanation methods to provide a relatively faster explanation. However, gradient-based saliency methods still need extra time to perform the sampling process on each instance [1,2]. The explanation derivation time is still highly dependent on the number of sampling and tested instances. RTX provides the model explanation in a very short time via a single feed-forward process (i.e., the derivation time does not increase drastically when the number of instances rises). Despite the differences between the gradient-based saliency methods and RTX, we have added the discussion in the related work highlighted in Appendix F. The following table shows the executing time of gradient-based saliency methods and real-time explainers. The results indicate that a real-time explainer is much faster than gradient-based saliency methods during the derivation process.
>
> | Method | IG | SmoothGrad | CoRTX |
> | :----: | :----: | :----: | :----: |
> | Time (**ms per image**) | 54.6 | 60.0 | 0.4 |
>
> [1] Smilkov, Daniel, et al. "Smoothgrad: removing noise by adding noise." arXiv preprint arXiv:1706.03825 (2017).
>
> [2] Sundararajan, Mukund, Ankur Taly, and Qiqi Yan. "Axiomatic attribution for deep networks." International conference on machine learning. PMLR, 2017.
>
>
> **Q2: I would prefer more intuition/explanation of how contrastive learning, especially the instances selection strategy, is useful in learning effective representation for the explainer.**
>
> [A2]: Thanks for your comment. The augmented positive and negative instances of explanation are proved to be beneficial to model explanation[1,2]. This is the reason we propose the explanation-oriented data augmentation strategy to learn explanation representation. In our work, Theorem 1 shows that our proposed positive instance selection strategy is able to select a positive explanation sample of the given anchor point. Based on the effects of contrastive learning[3], the representation of the given anchor point is similar to the representation of a positive explanation sample and dissimilar to the representation of negative explanation samples. This makes the generated representations contain the information of model explanation due to the selected positive pair in contrastive learning. We further provide Theorem 2 to show CoRTX can provide effective model explanations after fine-tuning based on the learned explanation representation.
>
> [1] Kim, Been, Rajiv Khanna, and Oluwasanmi O. Koyejo. "Examples are not enough, learn to criticize! criticism for interpretability." Advances in neural information processing systems 29 (2016).
>
> [2] Dhurandhar, Amit, et al. "Explanations based on the missing: Towards contrastive explanations with pertinent negatives." Advances in neural information processing systems 31 (2018).
>
> [3] He, Kaiming, et al. "Momentum contrast for unsupervised visual representation learning." Proceedings of the IEEE/CVF conference on computer vision and pattern recognition. 2020.

---

> > ### Comment · Reviewer_A6qm · 2022-12-14
> > **Response to rebuttal**
> >
> > Thank you for your detailed answers to my questions.
> > The score is raised in consideration of your answers.

---

> ### Author Response · Authors · 2022-11-13
> **Response to Reviewer A6qm. [Part 2/2 Q3 Q4 Q5]**
>
> **Q3: In the abstract, the authors claimed that “accurate explanation labels are hard to obtain due to limited human efforts.” Is this true?**
>
> [A3]: Thanks for your comment. We believe the statement is true. For example, in the healthcare domain, generating accurate explanation labels and results requires a lot of effort from domain experts. The evaluation of XAI in the healthcare domain is very challenging since it requires lots of labeling efforts from experts [1].
> One of the works of XAI on healthcare[2] proves that existing XAI methods are insufficient to fulfill the clinical requirements endorsed by doctors’ judgments. This work further proposes a metric based on clinical patterns from physicians as the ground truth scores to qualitatively evaluate the explanation results. The evaluation metrics are designed based on the physicians’ assessment to ensure efficacy. To sum up, evaluating XAI in the healthcare domain is difficult since the explanation labels are hard to acquire and require support from domain experts.
>
> [1] Ras, Gabriëlle, Marcel van Gerven, and Pim Haselager. "Explanation methods in deep learning: Users, values, concerns and challenges." Explainable and interpretable models in computer vision and machine learning. Springer, Cham, 2018. 19-36.
>
> [2] Jin, Weina, Xiaoxiao Li, and Ghassan Hamarneh. "Evaluating Explainable AI on a Multi-Modal Medical Imaging Task: Can Existing Algorithms Fulfill Clinical Requirements?." Association for the Advancement of Artificial Intelligence Conference (AAAI), volume 000. 2022.
>
>
> **Q4: In Figure 5, it would be useful to include the result of SHAP.**
>
> [A4]:  Thanks for your comment. We would like to clarify that SHAP has been one of the baselines compared with CoRTX shown in the experiments. In this work, SHAP  with low perturbation sampling is noted as KernelSHAP (KS) in Figure 5.
>
> **Q5: In Section 4.2, the equation to obtain hat{r} is not clear.**
>
> [A5]: Thank you for pointing out this part. We would like to use a following notation table to further describe the notations used in Feature Importance Ranking Task (Section 4.2). We have added the notations to be more clear in our revised version in Appendix J.
>
> | Notation | Definition \& Description | Dimension |
> | :----: | :----: | :----: |
> | $\hat{\mathbf{r}}_j$ | Predicted ranking index score list of feature $x_j$ from CoRTX. Each index score in $\hat{\mathbf{r}}_j$ represents the score of feature $x_j$ that is ranked at this index. | $\hat{\mathbf{r}}_j \in \mathbb{R}^M$| |
> | $\hat{\mathrm{r}}_j$ | Predicted ranking index from CoRTX, where $\hat{\mathrm{r}}_j = \arg \max \hat{\mathbf{r}}_j$ for $1 \leq j \leq M$ | $\hat{\mathrm{r}}_j \in \mathbb{R}^M$ |
> | $\mathrm{r}_j$ | The ground-truth ranking label sorted from Shapley values. | $\mathrm{r}_j \in \mathbb{R}^M$  |
>
> Here, we provide a toy example. We assume M=2, $\hat{\mathbf{r}}_1 = [0.2, 0.8]$, and $\hat{\mathbf{r}}_2 = [0.7, 0.3]$. $\hat{\mathbf{r}}_1$ indicates the ranking probability of feature 1, which 0.2 refers to the probability of ranking at first place and 0.8 refers to the probability of ranking at second place. Here, we have $\hat{\mathrm{r}}_1 = \arg\max \hat{\mathbf{r}}_1 = 2$ and $\hat{\mathrm{r}}_2 = \arg\max \hat{\mathbf{r}}_2 = 1$, which means that the feature importance ranking is [feature 2, feature 1].

---

> ### Author Response · Authors · 2022-11-18
> **To reviewer A6qm**
>
> Dear reviewer A6qm,
>
> Thanks again for your valued comments on our work. We have responded to your initial comments. As the deadline for context revising (Nov. 18th) is coming, we are looking forward to your feedback and will be happy to answer any further questions you may have.
>
> Sincerely,
>
> Authors

---

### Official Review · Reviewer_MLHw · 2022-10-26

**Confidence:** 3
**Correctness:** 2
**Technical Novelty And Significance:** 2
**Empirical Novelty And Significance:** 1
**Recommendation:** 6

**Clarity, Quality, Novelty And Reproducibility:**

Clarity---Good:

The paper is well written


Quality--Fair:

Some of the assumptions and experiments are not well explained. Please see the weakness section above.

Novelty--Medium:

The idea of estimating Shapley values is not novel it has been done before. The idea of using contrastive learning and a subset of labels for the estimation is novel.


Reproducibility --Good:

The code was provided and experimental details were clearly stated in the appendix which can enable the reproduction of the results.

**Strength And Weaknesses:**

Strength:
--
- The paper is well-written and easy to follow.
- The need for a fast reliable explanation method is crucial.

Weakness:
--
- I am not convinced this framework is telling us anything about how a model is making a prediction, the only time the model in question is actually used to find the positive samples in the contrastive loss. Why are the produced representations "Explanation representation"  unless I am missing something the produced representations are simply just different representations. And until they are fine-tuned with ground truth explanations they have nothing to do with model explanations.  The FASTShap paper was sold as a faster way to calculate an estimate of the Shapley values of a model. This might be the case here but claiming that the representation produced by the encoder is an explanation representation is not well supported.
- I strongly disagree with the notation of the ground truth explanations, especially when using an approximation of Shapley values as ground truth as done in two of the datasets in the experimental section.
- The "Supervised RTX" or "SOTA RTX" as explained in the appendix as "A supervised RTX-based MLP model trains with raw features of data instances and ground-truth explanation labels from scratch" Where was this introduced? Which paper I could not find a reference for?
- Why was there inconsistency when choosing ground truth I understand that Shapley values are expensive to calculate but why APS on one dataset and KS on another?

- Experiments:
    -  For "EXPLANATION EFFICACY AND EFFICIENCY": I find it very strange to be comparing with methods (PS/KS) that the paper has considered as ground-truth explanations. Is the rank accuracy and error reported against the actual Shapley values?
    - For "CONTRIBUTIONS ON EXPLANATION REPRESENTATION" The paper is mainly comparing with Supervised RTX which again I am not sure which paper proposes this method (if this method has not been previously proposed by another paper then basically, this is a made-up baseline, and not realistic).
    -  For "QUANTITATIVE EVALUATION" how is the masking and accuracy drop measured do you check the drop on the existing method or create an evaluation model similar to (Jethani et al., 2021)?


**Summary Of The Paper:**

The paper proposes a contrastive real-time explanation framework CoRTX, it is a framework based on contrastive learning to learn model explanations using contrastive learning and fine-tuning a corresponding explanation head using a small amount of ground truth explanation labels. Here the paper used Shapley value as the ground truth.
The contrastive framework consists of the following given:
(a) Train an explanation encoder to produce an explanation representation
 - Given a sample $x$, a set of synthetic positive instances  $X$ is generated via perturbations on x given by equation 2.
 - From set $X$ select $\tilde{x}^+$ that minimizes equation 4. So $\tilde{x}^+$ is the synthetic sample that gives the closet function output to $x$ when given to the target model.
-  Select a set of samples from the training dataset negative samples (any samples other than the original one x).
- Train the encoder by minimizing the typical contrastive loss in equation 5.
(b) When the encoder is trained, an explanation head is added which is a model that takes as an input the output of the encoder and produces the Shapley values as an output. This entire architecture is then finetuned with Shapley value as ground truth labels.

Two versions of CoRTX were introduced CoRTX-MSE where the model was trained to minimize the L2 distance and CoRTX-CE where the model is trained to minimize the cross entropy of the ranking.

The paper performed experiments on two tabular datasets Census and Bankruptcy, and one image dataset CIFAR-10. For ground truth explanations the paper used Shapley values for Census dataset, Antithetical Permutation Sampling for Bankruptcy and kernalShap for  CIFAR-10 as an approximation of Shapley values.

**Summary Of The Review:**

Overall, the idea of estimating Shapley values is important.
My main concerns in the paper are as follows:

- Why is representation produced by the encoder an "explanation representation"?
- Using an approximation of Shapley values as ground truth if Shapley values are too expensive to compute that even practically computing 5% of a large training dataset like imagenet is unfeasible then the overall idea of this paper is not valid but using an estimation of Shapley values to estimate the Shapley values seems incorrect.
- The paper mainly compares this idea of supervised RTX which is a model trained on all Shapley values and produces an explanation, however, this model does not really exist (please correct me if I am wrong and cite the paper accordingly) it is a made-up baseline introduced by the paper. Because as mentioned in the paper the ground-truth explanations are too expensive to generate in general.
- Some of the experimental section are unclear please see the weakness section.

---

> ### Author Response · Authors · 2022-11-13
> **Response to Reviewer MLHw. [Part 1/3 Q1 Q2]**
>
> We thank the reviewer for the constructive comments.
>
> **Q1: Why is representation produced by the encoder an "explanation representation"? Claiming faster way to calculate feature attribution might be the case in this paper but claiming the representation produced by the encoder is an explanation representation is not well supported.**
>
> [A1]: Thanks for your constructive comment. We are sorry about the confusion caused by the term “explanation representation.” Explanation representation is the latent representation produced from contrastive learning; however, the latent representation in CoRTX is encoded with the explanation information due to a positive instance selection strategy. We have rephrased the term “explanation representation” into “latent explanation” to prevent further misunderstanding. The latent explanation is defined as the encoded vector being helpful to derive the model explanation, in which the vectors contain explanation information. The modified parts are highlighted in blue in the main context.
>
> We respectfully disagree with the statement that the latent explanation encoded with explanation information is not well supported. The supportive evidence is from Theorem 1 and Theorem 2, which show that latent explanation obtains explanation information. Several prior works [1,2] prove that positive and negative examples of explanation are beneficial to understand the model explanation, which is the reason why the explanation-oriented data augmentation strategy is applicable to train latent explanation. In our paper, Theorem 1 shows that the positive pair chosen from the positive instance selection strategy obtains a similar model explanation to the given anchor point. According to the effect of contrastive learning[3], the representation of the given anchor point is similar to the representation of a positive explanation instance and dissimilar to the representation of negative explanation instances. The generated latent explanation is encoded with the model explanation information due to contrastive learning and selected positive pair. Theorem 2 further indicates that CoRTX can derive an effective model explanation after fine-tuning with the learned latent representation of explanation.
>
> [1] Kim, Been, Rajiv Khanna, and Oluwasanmi O. Koyejo. "Examples are not enough, learn to criticize! criticism for interpretability." Advances in neural information processing systems 29 (2016).
>
> [2] Dhurandhar, Amit, et al. "Explanations based on the missing: Towards contrastive explanations with pertinent negatives." Advances in neural information processing systems 31 (2018).
>
> [3] He, Kaiming, et al. "Momentum contrast for unsupervised visual representation learning." Proceedings of the IEEE/CVF conference on computer vision and pattern recognition. 2020.
>
>
> **Q2: Using the approximation of Shapley values as label is infeasible on large dataset such as Imagenet.**
>
> [A2]: Thanks for your comment. We agree with the reviewer that the label-generating time in the large-scale dataset is longer than in the small dataset. In this work, the proposed CoRTX only requires 5% of the training labels, significantly reducing the dependency on supervised labels and the time to generate labels. Yielding 5% of labels significantly reduces the computing time to generating 100% of labels, regardless of large-scale or small-scale datasets. The following table refers to the computational time for generating labels of 60,000 images on a single CPU device. The experiments follow the same settings from this work while generating the labels for the image dataset. The results show that 5% of label generation is significantly faster than 100% of label generation.
>
> | Proportion | 5%  | 10% | 50% |
> | :----: | :----: | :----: |  :----: |
> | Time (Hour) |  ~52.4 Hours | ~100.3 Hours | ~520 Hours |

---

> ### Author Response · Authors · 2022-11-13
> **Response to Reviewer MLHw. [Part 2/3 Q3, Q4, Q5]**
>
> **Q3: Reviewer disagrees with using approximation of Shapley values as ground truth.**
>
> [A3]: Thanks for your comment. We partially agree with the statement that the approximated Shapley value is not the exact Shapley value. However, using the high permutation times is acceptable in empirical experiments, especially for large-scale datasets. For example, FastSHAP[1] adopts the approximated Shapley values from KernelSHAP as the ground truth to evaluate the performance of the model explanation. Moreover, both KS[2] and APS[3] state and prove that the estimated values from KS and APS are able to approximate the exact Shapley values well when the permutation is high. Here, we conduct an experiment on the Adult dataset (13 features) to show that approximated values from KS and APS are very close to ground truth Shapley values when the permutation times are high. We evaluate KS and APS with 4096 perturbation times, shown in the following table. The results show the tiny errors between KS/APS and exact Shapley values, which makes it plausible to use approximated Shapley values as ground truth.
>
> [1] Jethani, Neil, et al. "FastSHAP: Real-Time Shapley Value Estimation." International Conference on Learning Representations. 2021.
>
> [2] Lundberg, Scott M., and Su-In Lee. "A unified approach to interpreting model predictions." Advances in neural information processing systems 30 (2017).
>
> [3] Mitchell, Rory, et al. "Sampling permutations for Shapley value estimation." (2022): 1-46.
>
> | Method | KS | APS |
> | :----: | :----: | :----: |
> | $\boldsymbol\ell_2\text{-error}$ | 0.0027 | 0.00024 |
>
>
> **Q4: The paper compares with supervised RTX. However, this model does not really exist, it is a made-up baseline introduced by the paper.**
>
> [A4]: Thanks for your comment. We would like to clarify that “supervised RTX” is only a naive approach to RTX in the supervised learning paradigm, which aims to prove that CoRTX is able to use fewer explanation labels during the training stage. The purpose of providing “supervised RTX” here is to show the limitations of RTX in supervised learning paradigm and highlight the advantages of CoRTX.
> Several advancements of supervised RTX has been proposed to show good explanation results. For example, FastSHAP[1] learns the distribution of approximated explanation labels by a DNN explainer, and it is equivalent to “supervised RTX” when the feature permutation times increase to infinity.
>
> [1] Jethani, Neil, et al. "FastSHAP: Real-Time Shapley Value Estimation." International Conference on Learning Representations. 2021.
>
>
> **Q5: Why was there inconsistency when choosing ground truth. I understand that Shapley values are expensive to calculate but why APS on one dataset and KS on another?**
>
> [A5]: Thanks for this comment. APS and KS are both good at approximating Shapley values when the permutation times are high. The choosing strategy of ground truth in this work depends on the properties of APS and KS. According to the following experiments on the adult dataset, it shows APS is more closed to the exact Shapley values on the tabular data. This is the reason we choose APS as the ground truth for tabular datasets. However, APS requires to estimate the importance score for each feature independently, which makes a burden for image data as its large-scale pixel numbers. Instead, KS generates the importance scores of all pixels simultaneously, which is the reason we adopt it on the image data.
>
> | Method | KS | APS |
> | :----: | :----: | :----: |
> | $\boldsymbol\ell_2\text{-error}$ | 0.0027 | 0.00024 |

---

> ### Author Response · Authors · 2022-11-13
> **Response to Reviewer MLHw. [Part 3/3 Q6, Q7]**
>
> **Q6: For "EXPLANATION EFFICACY AND EFFICIENCY": I find it very strange to be comparing with methods (PS/KS) that the paper has considered as ground-truth explanations. Is the rank accuracy and error reported against the actual Shapley values?**
>
> [A6]: Thanks for pointing out this question. The reported rank accuracy is evaluated by exact Shapley values on the Adult dataset, and the reported rank accuracy is evaluated by approximated Shapley values on the Bankruptcy dataset since the Bankruptcy dataset obtains large numbers of features.
> We use the outputs from PS/KS with very high permutation times as the ground truth (e.g., around $2 \times 10^5$ times for APS to generate ground truth for the Bankruptcy dataset). In the experiments, PS and KS, as the baselines, use relatively low permutation times, such as $2^3$ to $2^{10}$ for PS on the Bankruptcy dataset.
>
>
> **Q7: For "QUANTITATIVE EVALUATION" how is the masking and accuracy drop measured? Do you check the drop on the existing method or create an evaluation model similar to (Jethani et al., 2021)?**
>
> [A7]: Thanks for pointing out this question. The measurement of accuracy drop is based on the original prediction model (i.e., ResNet18 in our case) rather than a surrogate model as [1,2]. The reason why [1,2] adopts the surrogate models somehow solves the out-of-distribution (OOD) issues[5]. However, the surrogate model may overlook faithfulness since the targeted explaining model is no longer the original one but the surrogate model.
> In this manner, most of the prior work[3,4] chooses to exploit the original prediction model instead of training a new surrogate model. We are aware of the OOD problem from the original prediction model in the evaluation tasks. However, it is challenging to achieve both settings on faithfulness and OOD prevention perfectly. Thus, we choose to follow the common settings[3,4] that remain the faithfulness of derived explanation.
>
>
> [1] Jethani, Neil, et al. "FastSHAP: Real-Time Shapley Value Estimation." International Conference on Learning Representations. 2021.
>
> [2] Jethani, Neil, et al. "Have We Learned to Explain?: How Interpretability Methods Can Learn to Encode Predictions in their Interpretations." International Conference on Artificial Intelligence and Statistics. PMLR, 2021.
>
> [3] Petsiuk, Vitali, Abir Das, and Kate Saenko. "Rise: Randomized input sampling for explanation of black-box models." arXiv preprint arXiv:1806.07421 (2018).
>
> [4] Du, Mengnan, et al. "On attribution of recurrent neural network predictions via additive decomposition." The World Wide Web Conference. 2019.
>
> [5] Hooker, Sara, et al. "A benchmark for interpretability methods in deep neural networks." Advances in neural information processing systems 32 (2019).

---

> ### Author Response · Authors · 2022-11-18
> **To reviewer MLHw**
>
> Dear reviewer MLHw,
>
> Thanks again for your valued comments on our work. We have responded to your initial comments. As the deadline of context revising (Nov. 18th) is coming, we are looking forward to your feedback and will be happy to answer any further questions you may have.
>
> Sincerely,
>
> Authors

---

> > ### Comment · Reviewer_MLHw · 2022-11-21
> > **Thank you**
> >
> > I would like to thank the authors for their very detailed responses to all my questions.
> >
> > Q1, Q2 -> Addressed.
> >
> > Q3/Q6-> If this is the case then we should use  PS/KS with high permutation and that's it. You need to give a very very good argument against using high permutated PS/KS, like for example benchmark against different numbers of permutations and show time for each and compare that with the proposed method. Currently, I am not convinced.
> >
> > Q4->In this case you need to replace it with a more realistic baseline that is used in other papers.
> >
> > Q5-> Thank you please clarify this in the final draft.
> >
> > Q7-> Given the OOD problem I think the results are questionable, I do understand your argument but you can at least show the results for both. Since for me personally, I believe the OOD problem is a much more impactful issue than that introduced by surrogate models. Another option is to apply ROAR [1] approach.
> >
> >
> > [1] Hooker, Sara, et al. "A benchmark for interpretability methods in deep neural networks." Advances in neural information processing systems 32 (2019)
> >
> >
> > --------------------------
> > Overall:
> > I think the authors did a great job in the rebuttal but some fundamental problems (PS/KS vs CoRTX, weak baseline and evaluation OOD) remain questionable. For this reason, my score remains unchanged.

---

> > > ### Author Response · Authors · 2022-11-22
> > > **Follow-up Response to Reviewer MLHw. [ Part 1/2 FQ-1, FQ-2]**
> > >
> > > Thanks to the reviewer for the constructive comments to improve the quality of this work. We would like to further answer the follow-up questions raised by the reviewer. The additional experiment results will be included in the main context.
> > >
> > > **FQ-1: If this is the case then we should use PS/KS with high permutation and that's it. You need to give a very very good argument against using high permutated PS/KS, like for example benchmark against different numbers of permutations and show time for each and compare that with the proposed method.**
> > >
> > > [FA-1]: Thanks for your further comments. We are unsure of the part "using PS/KS with high permutation" you mentioned is for the ground truth generation (Q3) or the baseline settings (Q6). Thus, we clarify the question on both sides.
> > >
> > > - If the case is for the ground truth generation (Q3), we conducted additional experiments on the Adult dataset to show that PS/KS with high permutation is very similar to the exact values of Shapley values, especially when the perturbation is above 4000 times. This shows that high perturbated KS/PS is sufficient to generate the ground truth. There is no need to do higher permutation sampling for KS/PS to generate the ground truth. This is because the error remains extremely small as well, but the execution time grows drastically.
> > >
> > > | # of permutation | 2000  | 4000 | 6000 | 8000 |
> > > | :---- | :----: | :----: |  :----: |  :----: |
> > > | KernelSHAP ($\boldsymbol\ell_2\text{-error}$) | 0.0038 | 0.0027 | 0.0021 | 0.0018  |
> > > | APS ($\boldsymbol\ell_2\text{-error}$) | 0.00031 | 0.00025 | 0.00020 | 0.00017    |
> > > | KernelSHAP (sec/instance) | 0.3351 | 0.6816 | 0.9869 | 1.3089 |
> > > | APS (sec/instance)  | 0.0621 | 0.1164  | 0.1705 | 0.2222 |
> > >
> > > - If the case is for the baseline settings (Q6), we agree with the reviewer that PS/KS with very high permutation can be directly used as baselines. However, **comparing CoRTX with high-permutated KS/PS is unfair**, because the exceptionally high execution time of high-permutated KS/PS cannot be ignored in the experiments.
> > > Specifically, we conduct an experiment on the Adult dataset, and indicate the explanation performance and the execution time of PS/KS and CoRTX in the following table. It is observed that there is a trade-off between the speed and performance of KS/PS. **A fair comparison should keep the execution time as close as possible.** According to the results, CoRTX generates the model explanation in around 0.00015 seconds per instance, while high-permutated PS/KS requires at least 2000x longer execution time. Therefore, we believe that **comparing CoRTX with low-permutated KS/PS is more fair than with high-permutated KS/PS**. All experimental settings follow FastSHAP [1].
> > >
> > > | KS-(# of perturbation)| KS-300 | KS-2000 | KS-4000 | KS-6000 | KS-8000 | CoRTX-MSE |
> > > | :---- | :----: | :----: | :----: |  :----: |  :----: | :----: |
> > > | $\boldsymbol\ell_2\text{-error}$ | 0.0189 | 0.0038 | 0.0027 | 0.0021 | 0.0018  | 0.007 |
> > > | Time (sec/instance) | 0.1477 | 0.3351 | 0.6816 | 0.9869 | 1.3089 | **0.00015** |
> > >
> > >
> > > | APS-(# of perturbation) | APS-300 | APS-2000 | APS-4000 | APS-6000 | APS-8000 | CoRTX-MSE |
> > > | :---- | :----: | :----: | :----: |  :----: |  :----: | :----: |
> > > | $\boldsymbol\ell_2\text{-error}$ | 0.0171 | 0.00031 | 0.00025 | 0.00020 | 0.00017 | 0.007 |
> > > | Time (sec/instance) | 0.0380 | 0.0621 | 0.1164  | 0.1705 | 0.2222 | **0.00015** |
> > >
> > > [1] Jethani, Neil, et al. "FastSHAP: Real-Time Shapley Value Estimation." International Conference on Learning Representations. 2021.
> > >
> > > **FQ-2: In this case you need to replace it with a more realistic baseline that is used in other papers.**
> > >
> > > [FA-2]: Thanks for your further comments. We sincerely value the words of the reviewer. Following the comments from the reviewer, we added one more realistic RTX baseline, the L2X model [1], to compare with CoRTX on the Adult dataset. The additional experimental results are shown in the following table, indicating that CoRTX outperforms two kinds of realistic RTX baselines. We would like to clarify that we have already compared several realistic baselines with CoRTX in our paper, such as FastSHAP [2], a SOTA realistic baseline of supervised RTX, and other realistic non-amortized baselines (e.g., KS and PS).
> > >
> > > | Methods | L2X [1] | FastSHAP [2] | CoRTX-MSE | CoRTX-CE |
> > > | :----: | :----: | :----: | :----: | :----: |
> > > | Rank ACC | 0.0847 $\pm$ 0.0011 | 0.5485 $\pm$ 0.0040 | 0.6368 $\pm$ 0.0067 | **0.7680 $\pm$ 0.0031** |
> > >
> > > [1] Chen, Jianbo, et al. "Learning to explain: An information-theoretic perspective on model interpretation." International Conference on Machine Learning. PMLR, 2018.
> > >
> > > [2] Jethani, Neil, et al. "FastSHAP: Real-Time Shapley Value Estimation." International Conference on Learning Representations. 2021.

---

> > > > ### Comment · Reviewer_MLHw · 2022-11-23
> > > > **Thanks**
> > > >
> > > > Thank you for the hard work and effort put in a very small time. It is very impressive.
> > > > Based on the new results I will raise my score. Please be sure to include these results in the final draft.

---

> > > > > ### Author Response · Authors · 2022-11-23
> > > > > **Thank you**
> > > > >
> > > > > Thank you for your positive feedback and response. We thank the reviewer for the constructive comments to improve our paper. We will definitely include the additional experimental results in the final draft.
> > > > >
> > > > > Sincerely,
> > > > >
> > > > > Authors

---

> > > ### Author Response · Authors · 2022-11-22
> > > **Follow-up Response to Reviewer MLHw. [ Part 2/2 FQ-3]**
> > >
> > > **FQ-3: Given the OOD problem, I do understand your argument but you can at least show the results for both.**
> > >
> > > [FA-3]: Thanks for your further comments. We sincerely value the comments of the reviewer. In order to eliminate your concern, we conducted extra quantitative experiments by using the surrogate model, where the experiment settings follow from [1,2] mentioned by the reviewer. The following table shows the results of the exclusion and inclusion AUC under 1000 images. We observe that CoRTX outperforms other non-amortized baselines on both inclusion and exclusion tasks. CoRTX performs better than FastSHAP (a SOTA of supervised RTX method) on the exclusion task and is competitive with FastSHAP on the inclusion task. The results reveal a similar pattern as using the original prediction model in the quantitative experiments, which proves that CoRTX has the capability to derive effective model explanations on either of the experiment settings.
> > >
> > > | Top-1 Accuracy | Exclusion | Inclusion |
> > > | :----: | :----: | :----: |
> > > | CoRTX | **0.372 $\pm$ 0.012** | **0.772 $\pm$ 0.012** |
> > > | FastSHAP | 0.386 $\pm$ 0.012 | **0.767 $\pm$ 0.013** |
> > > | KernelSHAP | 0.406 $\pm$ 0.011 | **0.767 $\pm$ 0.012** |
> > > | Saliency | 0.543 $\pm$ 0.012 | 0.703 $\pm$ 0.012 |
> > > | IG | 0.581 $\pm$ 0.012 | 0.704 $\pm$ 0.012 |
> > > | Smoothgrad | 0.480 $\pm$ 0.012 | 0.695 $\pm$ 0.012 |
> > > | Gradcam | 0.573 $\pm$ 0.012 | 0.719 $\pm$ 0.013 |
> > > | Deepshap | 0.573 $\pm$ 0.012 | 0.706 $\pm$ 0.012 |
> > >
> > > Finally, we would like to clarify that our work focus on the **efficiency and efficacy** of model explanation.
> > > We thank the reviewer for the comment since OOD is an important problem in XAI tasks. We will focus on the OOD problem in our future work.
> > >
> > > [1] Jethani, Neil, et al. "FastSHAP: Real-Time Shapley Value Estimation." International Conference on Learning Representations. 2021.
> > >
> > > [2] Jethani, Neil, et al. "Have We Learned to Explain?: How Interpretability Methods Can Learn to Encode Predictions in their Interpretations." International Conference on Artificial Intelligence and Statistics. PMLR, 2021.

---

### Author Response · Authors · 2022-11-13
**General Comments for all reviewers.**

Dear reviewers,

We thank all reviewers for their constructive reviews. We have revised the paper accordingly and marked the modifications in blue for visibility.

We are grateful to all reviewers for their constructive comments and helpful feedback. We are pleased to find that they find our contribution novel (ZCbd, A6qm, MLHw), clearly written (ZCbd, A6qm, MLHw), and the experiments well established (ZCbd, A6qm, bBUe).

To address your primary concerns, we have done our best to modify the work. We explicitly explain the explanation-oriented positive pair in CoRTX enables the learned representation to contain the explanation information. The effects of the contrastive framework in CoRTX encode the explanation information into the learned representation. In the evaluation of explanation tasks, explanation labels are difficult to acquire. The learned representation can effectively reduce the required amounts of explanation labels during the fine-tuning phase.

The revision parts are summarized as follows:

- (MLHw, A6qm, bBUe) We have added the intuition of learned latent explanation by CoRTX in Section 3.2.

- (MLHw) We have rephrased the term “explanation representation” into “latent explanation” throughout the paper to prevent further confusion.

- (A6qm) We have added the additional table in Appendix $\mathrm{j}$ to illustrate the notations of Section 4.2.

- (bBUe) We have added additional experiments to compare with more baselines (L2X, RISE, and XRAI) in Appendix $\mathrm{I}$.

- (bBUe) We have revised the discussion of the case study on the image dataset in Section 4.4.1.

- (bBUe) We have improved the readability and clarity of Figures 2, 4, 5.

- (ZCbd) We have explicitly illustrates the explanation labels are hard to acquired in Section 2.2, which motivates our unsupervised learning paradigm of CoRTX.

We appreciate all the suggestions made by reviewers to improve our work. We are pleased to hear your feedback and look forward to answering your follow-up questions.

Sincerely,

Authors

---

### Author Response · Authors · 2022-12-06
**Rebuttal Summary**

# Summary of Rebuttal
We thank all the reviewers and AC for their efforts and time in evaluating our work. We are pleased to find that all of the reviewers appreciate our contribution to the novelty with well-supportive theorems and experiments in our work.
We sincerely value the constructive comments of the reviewers during the rebuttal session, specifically the encouraging feedback and active responses from the Reviewer MLHw, bBUe, and ZCbd. We are glad that the main concerns and follow-up questions from the Reviewer MLHw, bBUe, and ZCbd has been addressed during the rebuttal session. After re-evaluating the revised version, it is happy to see two active Reviewer MLHw and bBUe raise their evaluation scores. One active Reviewer ZCbd maintains a score of 8 according to the new version of our work.y

As for Review A6qm, we firmly believe that our initial response has addressed all of your concerns since there are no further comments raised from your side till now. We also rephrased your concerns in our main text, and we believe they are all significantly improved. Seeing that the discussion panel has come to an end, we are still looking forward to your further feedback. If our response has addressed your concerns, it is very kind of you to re-evaluate our paper based on the revised version as you commit.

===== Post response on 12/13 =====

We thank the effort and time of Reviewer A6qm for re-evaluating our work. We are glad our initial response solves your concerns, and the score is raised according to the revised version.

# Contributions of Our Work
According to the initial reviews and the follow-up feedback on the rebuttal, we are delighted that all of the reviewers appreciate our contributions. In this work, we propose the contrastive framework CoRTX to provide a real-time estimation of model explanation, which encodes the explanation information into the learned representation. In the evaluation of explanation tasks, explanation labels are difficult to acquire. With the learned latent representation, we can effectively reduce the reliability of explanation labels during the fine-tuning phase. CoRTX can thereby provide effective and real-time model explanations underlying prediction models when making decisions.

---

### Decision · Program_Chairs · 2023-01-20

**Decision:**

Accept: poster

**Justification For Why Not Higher Score:**

The paper is limited to shapley based explanation.

**Justification For Why Not Lower Score:**

This is a solid paper that provides an efficient and effective solution for generating real time shapley explanation.

**Metareview: Summary, Strengths And Weaknesses:**

This paper introduce CoRTX, a contrastive learning framework that can provide real-time estimation of Shapley value explanations without require . Prior work for real time shapley estimation relied on large number of ground truth shapley labels to ensure accurate estimation. This work reduces the dependence on the labXels by introducing a pretraining step to learn a latent explanation representation. It presents theoretical results establishing an upper bound on the explanation error, which motivates the design for positive instance selection and the representation learning loss.  Experimental results show that CoRTX is both effective and efficient against a large set of baselines.

Strength:
- Well motivated problem with a reasonable and effective solution.
- The method is theoretical motivated and justified.
- Strong results showing the effectiveness of the proposed method

Weakness:
- The method is limited to shapley based explanation

A few minor comments.
- Equation 2, S_i is sampled from M-dim binomial distribution, do you mean M-dim Bernoulli distribution?
- It is interesting to note that theorem 1 and 2 do not appear to depend on X^+, what is the implication of this?
- the paper needs a careful proofreading. e.g., 4.4.1 has a missing appendix reference

**Note From Pc:**

if the above contains the word "oral" or "spotlight" please see: "oral" presentation means -> notable-top-5% and "spotlight" means -> notable-top-25%. As stated in our emails, we are disassociating presentation type from AC recommendations

**Summary Of Ac-Reviewer Meeting:**

This was originally listed as a borderline paper but the reviewer reached consensus during the discussion/rebuttal period, hence removed from the borderline list.